# Major sex differences in allele frequencies for X chromosomal variants in both the 1000 Genomes Project and gnomAD

Zhong Wang[1], Lei Sun[2,3]*, Andrew D. Paterson[3,4,5]*

1 Department of Statistics and Data Science, Faculty of Science, National University of Singapore, Singapore, 2 Department of Statistic Sciences, Faculty of Arts and Science, University of Toronto, Ontario, Canada, 3 Biostatistics Division, Dalla Lana School of Public Health, University of Toronto, Ontario, Canada, 4 Genetics and Genome Biology, The Hospital for Sick Children, Ontario, Canada, 5 Epidemiology Division, Dalla Lana School of Public Health, University of Toronto, Ontario, Canada

* sun@utstat.toronto.edu (LS); andrew.paterson@sickkids.ca (ADP)

## Abstract

An unexpectedly high proportion of SNPs on the X chromosome in the 1000 Genomes Project phase 3 data were identified with significant sex differences in minor allele frequencies (sdMAF). sdMAF persisted for many of these SNPs in the recently released high coverage whole genome sequence of the 1000 Genomes Project that was aligned to GRCh38, and it was consistent between the five super-populations. Among the 245,825 common (MAF>5%) biallelic X-chromosomal SNPs in the phase 3 data presumed to be of high quality, 2,039 have genome-wide significant sdMAF (p-value <5e-8). sdMAF varied by location: non-pseudo-autosomal region (NPR) = 0.83%, pseudo-autosomal regions (PAR1) = 0.29%, PAR2 = 13.1%, and X-transposed region (XTR)/PAR3 = 0.85% of SNPs had sdMAF, and they were clustered at the NPR-PAR boundaries, among others. sdMAF at the NPR-PAR boundaries are biologically expected due to sex-linkage, but have generally been ignored in association studies. For comparison, similar analyses found only 6, 1 and 0 SNPs with significant sdMAF on chromosomes 1, 7 and 22, respectively. Similar sdMAF results for the X chromosome were obtained from the high coverage whole genome sequence data from gnomAD V 3.1.2 for both the non-Finnish European and African/African American samples. Future X chromosome analyses need to take sdMAF into account.

**Data Availability Statement:** All genotype data files are available from: Phase 3 data of the 1000 Genomes Project: http://ftp.1000genomes.ebi.ac.uk/vol1/ftp/release/20130502/ and the specific vcf

## Author summary

The human X chromosome contains over 800 genes and is the 8th largest human chromosome. Genome-wide associations studies have generally failed to examine variants on the X chromosome for association with diseases and traits, partly due to complexities of the data analysis, and challenges with genotype imputation. We examined X chromosomal variants from the 1000 Genomes Project for sex differences in allele frequency and found that many variants showed significant differences. These variants cluster at the centromeric parts of the pseudoautosomal regions 1 and 2, as well as the putative pseudo-

file used: ftp://ftp.1000genomes.ebi.ac.uk/vol1/ftp/release/20130502/ALL.chrX.phase3_shapeit2_mvncall_integrated_v1b.20130502.genotypes.vcf.gz High coverage data of the 1000 Genomes Project: http://ftp.1000genomes.ebi.ac.uk/vol1/ftp/data_collections/1000G_2504_high_coverage/working/20201028_3202_phased/CCDG_14151_B01_GRM_WGS_2020-08-05_chrX.filtered.eagle2-phased.v2.vcf.gz gnomAD v.3.1.2 allele counts from genome sequence: https://gnomad.broadinstitute.org/downloads chrX sites VCF: 95.6 GiB, MD5: 040080a18046533728fa60800eedcf4b gnomAD v.2.1 structural variants: (https://gnomad.broadinstitute.org/gene/ENSG00000124343?dataset=gnomad_sv_r2_1) code is here: https://github.com/ZhongWang99.

**Funding:** LS received the awards: This research is funded by the Natural Sciences and Engineering Research Council of Canada (NSERC, RGPIN-04934) and the Canadian Institutes of Health Research (CIHR, PJT-180460). The funders had no role in study design, data collection and analysis, decision to publish, or preparation of the manuscript.

autosomal region 3 (also termed X-transposed region). This pattern was observed in high coverage whole genome sequence data from the same subjects that was aligned to GRCh38, suggesting that is not an artefact of low coverage sequencing or problems specific to GRCh37. In addition, we replicated this phenomenon in high coverage whole genome sequence aligned to GRCh38 from the gnomAD database in both the non-Finnish European and African/African American samples. These findings have implications for the analysis of X chromosomal variants for disease and trait associations.

## Introduction

After the striking observation that the X chromosome was excluded from most genome-wide association studies (GWAS) [1], there has been a slow increase in the incorporation of analysis of the X chromosome [2–5]. Several association methods have recently been developed for the X chromosome [2–12], focussing on the phenomenon of X-inactivation [13], also known as dosage compensation. However, these X chromosome specific downstream methods, similar to those developed for the autosomes, typically presume high quality data and implicitly assume that there is no sex difference in minor allele frequency (sdMAF).

Most genotype calling, imputation and sequence analyses of X chromosomal variants apply methods and tools that were developed for the autosomes [1,2]. However, there are reports that genotype missing rate is higher for SNPs on the X chromosome than autosomes [1,2]. In the non-pseudoautosomal region (NPR) of the X chromosome males are hemizygous, meaning that the intensity of allele signals from genotyping arrays, or the number of reads from sequencing, is half that of homozygous females. This may result in variant positions having higher missing rates for males than females.

In addition to higher missing rates for variants on the X chromosome, the two pseudo-autosomal regions (PARs), PAR1 and PAR2, create further challenges for the analysis at the boundaries between PARs and NPR [14]. PAR1 is 2.75 Mb at the end of the short arms of the X and Y chromosomes (Xp22.33 and Yp11.32-p11.2) containing 16 genes, while PAR2 is at the tip of the long arms (Xq28 and Yq12) and is 320 kb, containing 4 genes [15–17]. Because there is no recombination of NPR in males, but recombination occurs in PAR1 and PAR2 in both sexes, variants in PARs close to the PAR-NPR boundaries are linked to variants in NPR of the X and Y chromosomes in a sex-specific fashion. Obligatory recombination occurs in PAR1 in males, making it the region with the highest recombination rate per physical distance in the human genome, while only ~2% of meiosis feature recombination at PAR2. Although the effects of sex-specific recombination rates in PAR1 and PAR2 on linkage have been examined for non-parametric linkage analysis of affected sibpairs [18], the implications for X chromosomal data collected for association studies have not been well explored. The XTR/PAR3 in Xq21.3/Yp11.1 adds further complexity as this 3.91 Mb region on the X chromosome (3.38 Mb on the Y chromosome) is embedded in the non-pseudoautosomal region and has 98.8% sequence homology between X and Y [19–22].

Recently, sex-differences in allele frequency in PAR1 and PAR2 were described using the African super-population from the phase 3 data of the 1000 Genomes Project [23], but the rest of the X chromosome and other four super-populations of the 1000 Genomes Project were not examined. Evolutionary dynamics, including recombination within the PAR regions, were reasoned as a major contributing factor to sdMAF, but genotyping errors and the agreement with the high coverage data were not examined.

We hypothesize that sex difference in MAF exists across the X chromosome in human populations, including NPR and at the boundaries between NPR and PAR1, PAR2 and PAR3, and it is more prevalent in low-coverage whole genome sequence data. We use publicly available phase 3 data from the 1000 Genomes Project to test for sdMAF across and within each of the five super-populations. We compare the results with the recently released high coverage whole genome sequence to determine whether genotyping error is contributory. Finally, we examined genotype data from the high coverage whole genome sequence gnomAD v3.1.2 resource to further evaluate sdMAF on the X chromosome in two defined populations with the largest sample sizes.

Better understanding of the different sources of sdMAF is critical to developing X chromosome-suitable analytical strategies, from improved data collection and imputation to more robust association methods for variants on the X chromosome.

## Results

### 1000 Genomes Project

The proportions of males and females were similar across the 26 populations of the 1000 Genomes Project (S1 Fig).

### Phase 3 X chromosome-wide sdMAF across super-populations (ALL)

For biallelic SNPs with global MAF≥5% (Fig 1 and S1 Table), the Manhattan plot for sdMAF p-values (Fig 2A) shows that a non-negligible proportion of X chromosomal SNPs have significant sex difference in MAF even at the genome-wide significance threshold of p-value <5e-8: 0.83% of SNPs in NPR, 0.29% in PAR1, 13.1% in PAR2, and 0.85% in PAR3. The excesses of small p-values for sdMAF testing in all four regions of the X chromosome are also evident from both the QQ plots (S2 Fig) and histograms of the sdMAF p-values (S3 Fig). SNPs with significant sdMAF are located across the X chromosome but tend to cluster in specific regions (Fig 2A).

### Super-population specific analyses

Our primary analysis (above) combined data from all five super-populations. We also performed analysis separately for each super-population to determine whether the effect magnitude and direction were consistent across super-populations. S4–S8 Figs present the results for each super-population and show that, based on the comparisons of the p values, there are generally similar sdMAF patterns across the five super-populations. The consistency is also present for the magnitude and direction of sdMAF (S9–S10 Figs).

The top row of S9 Fig shows that the ALL analysis is overall much more powerful than the super-population-specific analysis, identifying many SNPs with significant sdMAF that would be missed by the individual super-population analysis, and often the sdMAF p value is several magnitudes smaller.

In contrast, although the individual super-population (continent-stratified) analysis identified some SNPs with significant sdMAF that would be missed by the ALL analysis (S2 Table), the sdMAF p values are comparable as shown in S9 Fig. There is also a remarkable consistency in sdMAF estimates between the five super populations as shown in S10 Fig. We also performed a meta-analysis with sample-size based weighting of all five super-populations and compared the results to the primary ALL analysis (above). S11 Fig (Miami) and S12 Fig (pp plot) show that the results are consistent between the two analyses.

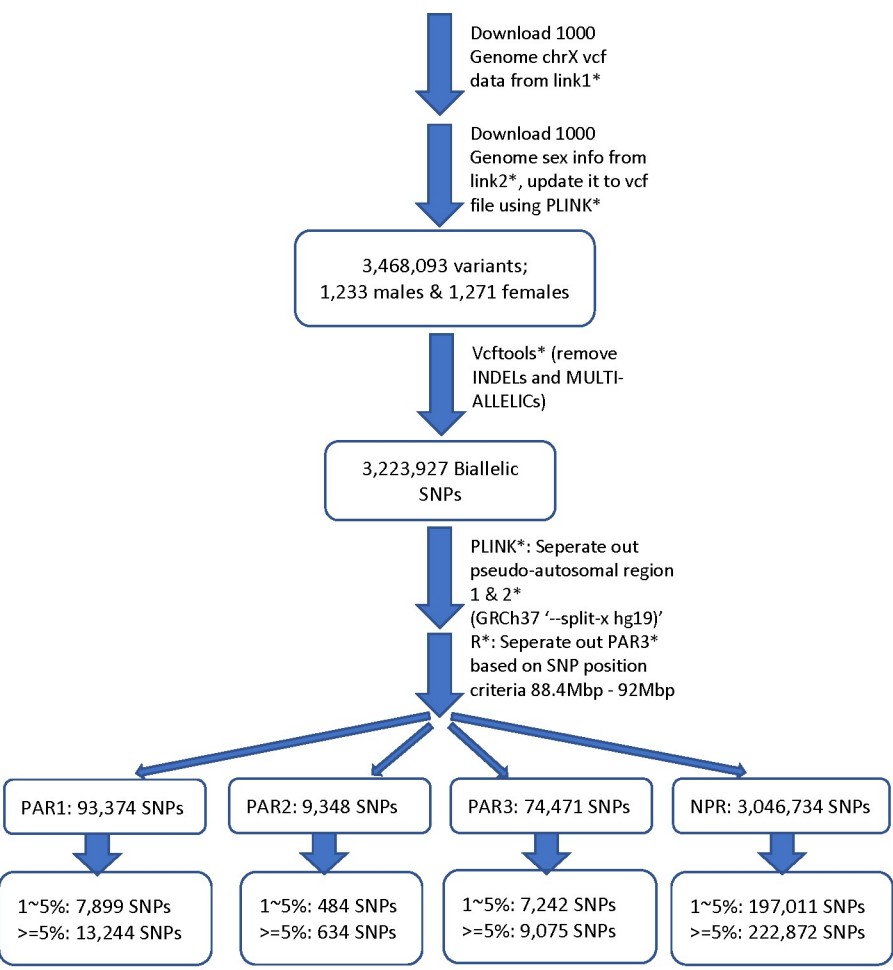

*link1: ftp://ftp-trace.ncbi.nih.gov/1000genomes/ftp/release/20130502/
ALL.chrX.phase3_shapeit2_mvncall_integrated_v1b.20130502.genotypes.vcf.gz
*link2: https://www.internationalgenome.org/data-portal/sample
*PLINK: 1.90 beta version 6.20 64-bit
*PAR1&PAR2 exact location
*vcftools: version 0.1.17
*R: version 3.5.3
*PAR3: No global consensus yet

**Fig 1. Pipeline for selection of X chromosome biallelic SNPs with global MAF ≥5%, presumed to be of high quality, from the 1000 Genomes Project phase 3 data on GRCh37.** Variants were placed into the NPR, PAR1, PAR2, and PAR3 regions based on positions available from The Genome Reference Consortium and [19]. For detailed counts of variant types and global MAF by regions, see S1 Table.

Although unlikely, it is possible that the above continent-stratified analysis may fail to identify sdMAF SNPs with heterogeneity (in terms of signs of the sdMAF estimates) in sdMAF between populations. Thus, for the eight X chromosomal SNPs with the most salient results (as initially discussed in S1 Data) from the 1000 Genomes phase 3 data, we further conducted a population-stratified sdMAF analysis, separately for each of the 26 populations.

Fig 3 shows that, as expected, the sdMAF results for these SNPs are consistent across the 26 populations. For example, for rs6634333 (POS = 140993859) from the NPR region (Fig 3H), the sdMAF estimate in the overall sample is 0.338 (p-value = 3.78E-151). While the 26

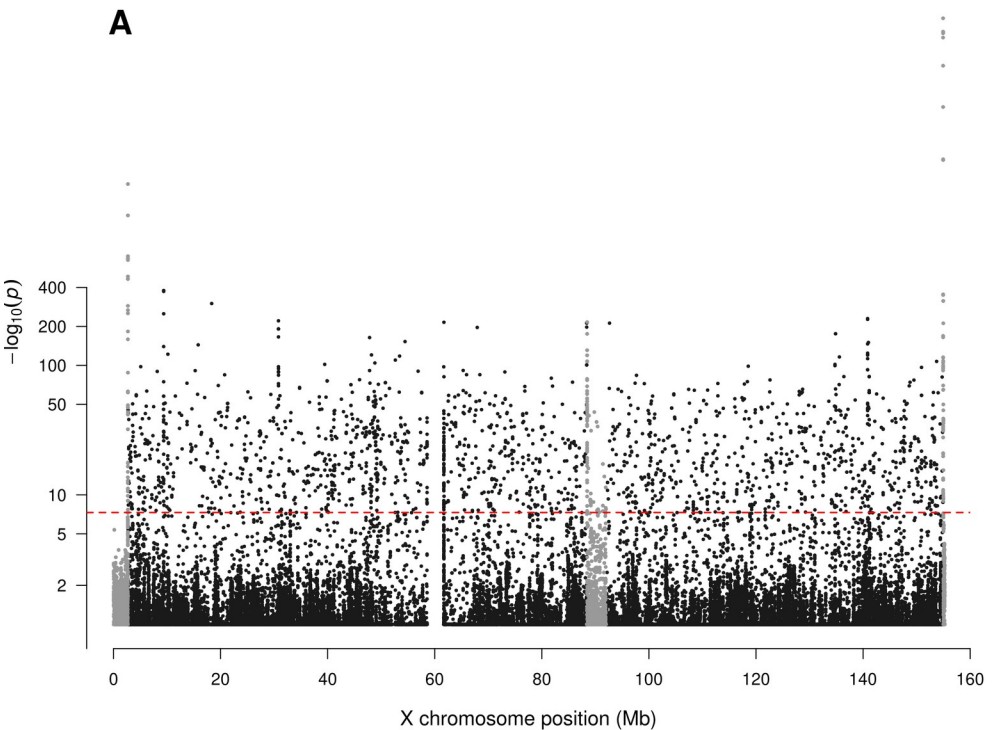

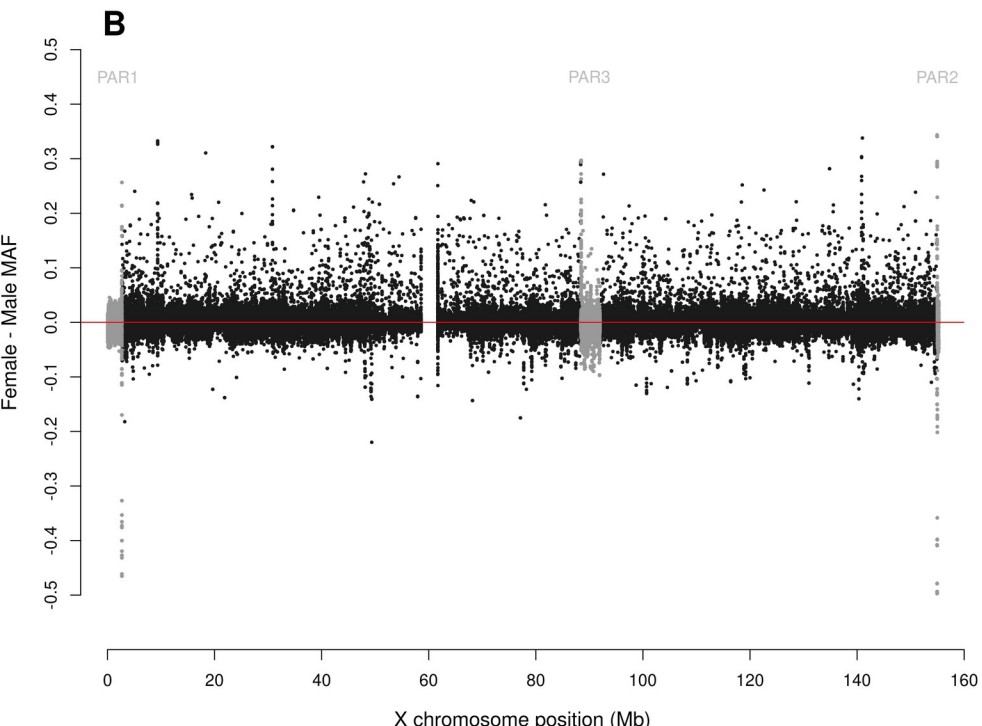

**Fig 2. Manhattan plot for testing for sex difference in MAF across the X chromosome from the 1000 Genomes Project phase 3 data on GRCh37.** A: sdMAF p-values for bi-allelic SNPs with global MAF ≥5% presumed to be of high quality. SNPs in the PAR1, PAR2 and PAR3 regions are plotted in grey, with PAR3 located around 90 Mb. Y-axis is −log10(sdMAF p-values) and p-values >0.1 are plotted as 0.1 (1 on −log10 scale) for better visualization. The dashed

red line represents 5e-8 (7.3 on the −log10 scale). B: Female—Male sdMAF for the same SNPs in part A. For Zoomed-in plots for the PAR1, PAR2 and PAR3 regions see Figs 4, 5 and 6, respectively.

population-specific estimates are not identical, all 26 estimates are positive and range from 0.206 (p-value = 0.011) in STU to 0.443 (p-value = 2.90E-9) in ESN (S5 Data). Remarkable consistencies across populations are also observed for the other seven SNPs (Fig 3).

## Regional variation in sdMAF

Fig 2B shows that the direction and magnitude of the sdMAF varies by genomic location. Among the analysed SNPs, 0.39%, 16.88%, 1.54% and 1.18% of the SNPs have absolute sdMAF $\geq$0.05, respectively in PAR1, PAR2, PAR3, and NPR. For these SNPs, the medians of absolute (sdMAF) are respectively 0.095, 0.121, 0.074, and 0.077, and the [Q1,Q3] are [0.060,0.274], [0.064,0.286], [0.060,0.099], and [0.059,0.111], respectively, in PAR1, PAR2, PAR3, and NPR regions. Of the 2,039 SNPs with significant sdMAF we observed females generally having higher MAF than males: NPR = 93%, PAR2 = 59% and PAR3 = 86% among the SNPs with genome-wide significant sdMAF, except for PAR1 = 31% (Figs 4–6). Specifically, females have higher MAF at around 30 Mb (GRCh37) and at the q-arm of the centromere (Fig 2B), as well as at the centromeric boundary of PAR3 (Fig 6). In contrast, at the region of PAR1 close to the NPR boundary, males tend to have higher MAF among the SNPs with significant sdMAF (Fig 4). Finally, there are sets of PAR2 SNPs close to the NPR boundary that have higher MAF in females, while other sets of SNPs have higher MAF in males (Fig 5).

We then examined how the sdMAF relates to the sex-combined MAF. Fig 6 provides Bland-Altman plots [24] separately for each of the four regions. Of note, for NPR and PAR3 (Fig 7A and 7D), SNPs with significant sdMAF tend to have higher MAFs in females and pre-dominantly had sex-combined MAFs in the range 25%-40%. In contrast, for PAR1 (Fig 7B), SNPs with significant sdMAF tend to have higher MAFs in males. Finally, for PAR2 (Fig 7C), there are sets of clustered SNPs with significant sdMAF in either direction (Fig 7C).

## In-depth analysis of eight SNPs with the most significant sdMAF in the phase 3 data

For the eight selected SNPs, two from each of the four regions (NPR, PAR1, PAR2, and PAR3) with the smallest sdMAF p-values in the combined sample, the population-specific sdMAF p-values remain genome-wide significant (S1 Data). Moreover, the directions of the sdMAF are consistent across the five super-populations. For example, for rs201194898 in NPR (GRCh37 position = 9,377,082), the sdMAF p-values are <1e-200, 8.82e-149, 1.12e-49, 2.50e-89, 2.91e-45, and 1.95e-101, respectively in the ALL (combined) and the EAS, EUR, AFR, AMR, and SAS super-populations; the corresponding female minus male sdMAFs are all >0.27. That is, the MAFs of rs201194898 are significantly larger in females than in males, across all five super-populations. Note that the minor allele, defined based on sex- and population-combined sample, may not have MAF less than 0.5 in a sex- and/or population-stratified sample, and for a SNP in NPR and PAR3 each male only contributes a single allele to the MAF calculation. Similar results are reported in S1 Data for rs6634333 in NPR, as well as the two PAR3 SNPs in the 88,460,295 and 88,462,611 positions (GRCh37).

For the PAR1 SNP at 2,697,599, the super-population sdMAF p-values are all <1e-200 (except 2.69e-80 in AMR); the corresponding female minus male sdMAFs are all more extreme than −0.46 (−0.34 in AMR). Similar results were observed for the PAR2 SNP at 154,934,295, for which sdMAF p-values <1e-200 in all five super-populations: MAFs are

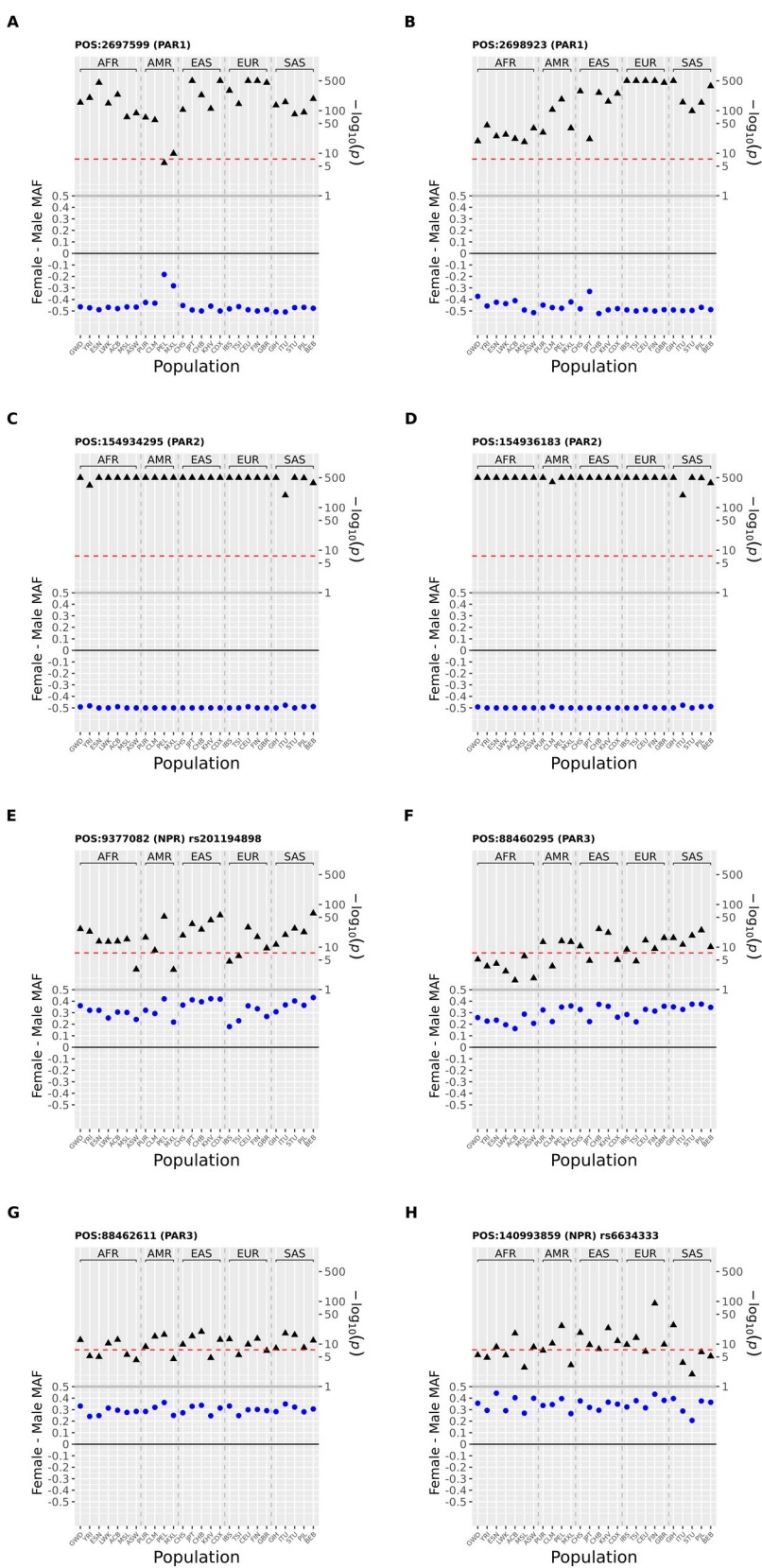

**Fig 3. Population-specific sdMAF results for 8 X chromosomal SNPs from 1000 Genomes Project phase 3 data across the 26 populations.** The 8 SNPs were selected (two each from the 4 regions: PAR1, PAR2, PAR3 and NPR) with the smallest sdMAF p values (see text). Analysis was performed separately for the 26 populations and sdMAF (female-male, blue circles) as well as their corresponding -$\log_{10}$(p value) (black triangles) are plotted on the left and right Y axes, respectively. Populations are labelled on the X axis using the conventional 1000 Genomes Project codes, and grouped by super-population. Physical position is GRCh37 and rs provided where available. The red dashed horizontal line is a p = 5x10$^{-8}$. SNPs in each part are: **A** position = 2697599 (PAR1); **B** position 2698923 (PAR1) and **C** position 154934295 (PAR2); **D** position 154936183 (PAR2); **E** position = 9377082 rs201194898 (NPR); **F** position = 88460295 (PAR3); **G** position = 88462611 (PAR3); **H** position = 140993859 rs6634333 (NPR).

significantly smaller in females than males. In addition, all females in EAS, EUR and AMR are homozygous TT, while males in EAS, AFR and AMR are all AT heterozygotes. In other words, genotype is fixed by sex in EAS and AMR.

The population-specific HWE testing results [25–27], however, vary across the eight SNPs or across the five super-populations. For example, for rs6634333 in NPR, the HWE testing p-values in females are genome-wide significant in all but SAS: 1.71e-30, 3.21e-34, 1.09e-18, 1.97–22, and 2.41e-4 respectively in EAS, EUR, AFR, AMR and SAS. The corresponding population-specific HWD delta estimates are all negative, with an excess of heterozygous females. For rs201194898 in NPR and the two PAR3 SNPs, population-stratified HWE testing in females are genome-wide significant, with excesses of heterozygous females in all five super-populations.

For the PAR1 SNP in the 2,697,599 position where HWE testing could be performed in both females and males, HWE testing in males are all genome-wide significant, but in females the HWE p-values >0.8; consistent results were observed for the other three SNPs in PAR1 and PAR2 (S1 Data). In addition, a SNP can be monomorphic in females (HWE p-value = NA in that case), while in the male sample the heterozygous genotype may be present for some of the five super-populations.

## Minor allele switch between males and females

Of the 2,039 SNPs with genome-wide significant sdMAF, 66 (3.24%) have minor allele flips between males and females. For this set of SNPs we provide S2 Data (similar in format to S1 Data).

Additionally, without requiring the sdMAF testing to be genome-wide significant, S13 Fig shows Bland-Altman plots, stratified by the four regions, for X chromosome SNPs with different minor alleles between males and females. Among this set of SNPs, only one SNP, in PAR2 (POS = 154941870), has close to sex-specific genotypes: A1A1 = 0, A1A2 = 20, A2A2 = 1251 in females, and A1A1 = 9, A1A2 = 1219, A2A2 = 5 in males, and has genotype-wide significant sdMAF (S2 Data).

## Sliding window analysis of phase 3 1000 Genomes X chromosome data

We performed a sliding window analysis, using a window size of 50 SNPs, sliding 25 SNPs at a time using the mean of the single SNP–log(10) p values. This identified ten regions across the X chromosome, of which the three at the boundaries of the PARs with NPR remain (S14 Fig).

## sdMAF in NPR homologs

There are 19 NPR genes with homologs on X and Y (S3 Data). Of the 2,039 SNPs with significant sdMAF, 52 were in homologs (S4 Data).

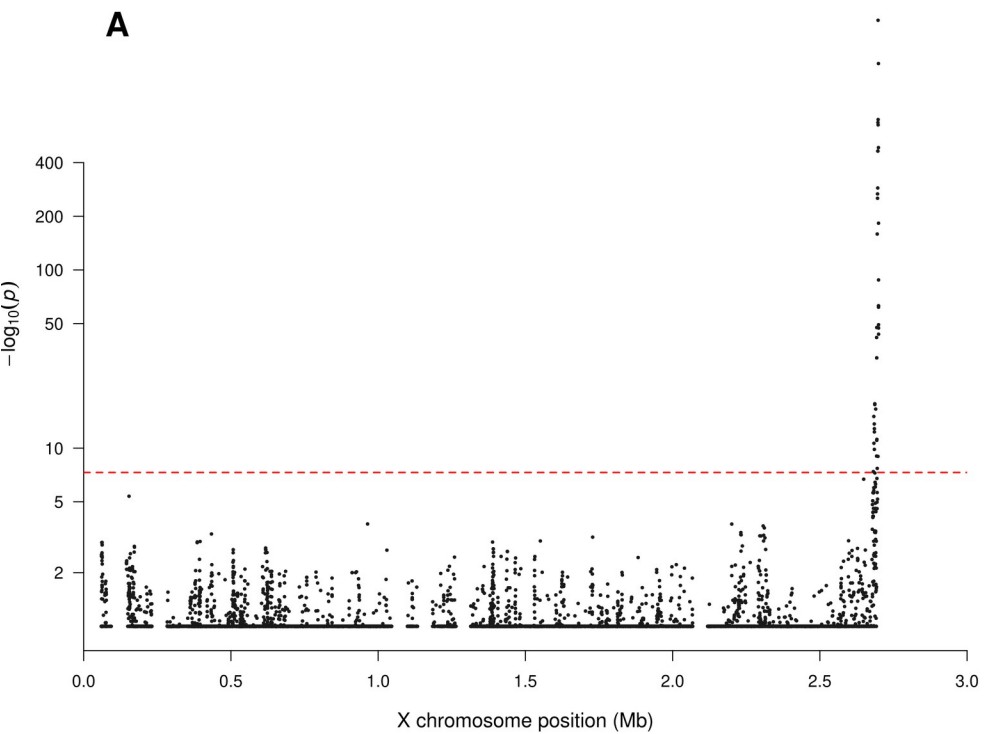

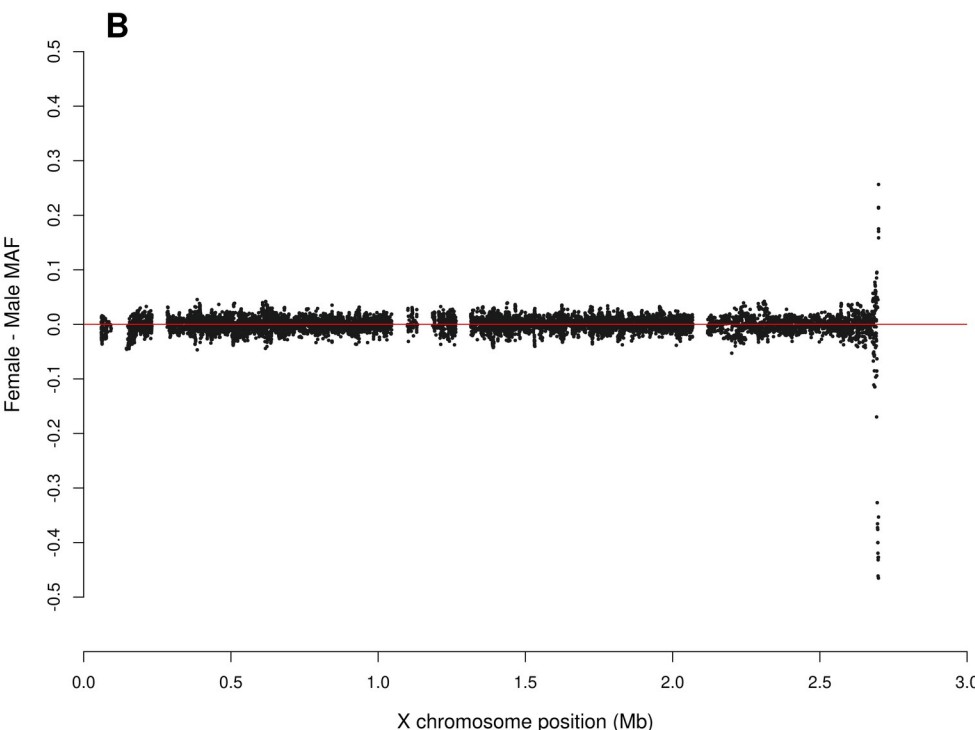

**Fig 4. Zoomed-in plot for testing for sex difference in MAF across PAR1 of the X chromosome from the 1000 Genomes Project phase 3 data on GRCh37.** A: sdMAF p-values for bi-allelic SNPs with global MAF ≥5% presumed to be of high quality. Y-axis is −log10(sdMAF p-values) and p-values >0.1 are plotted as 0.1 (1 on −log10 scale) for better visualization. The dashed red line represents 5e-8 (7.3 on the −log10 scale). B: Female—Male sdMAF for the

same SNPs in part A, clearly showing PAR1 SNPs with significant sdMAF tend to cluster at the NPR-PAR1 boundary around 2.6 Mb.

## Comparison to chromosomes 1, 7 and 22 in the 1000 Genomes Project phase 3 data

Chromosomes 1, 7 and 22 were selected as the longest, most similar in length to the X chromosome, and one of the shortest. There were 530,434, 406,057 and 97,216 biallelic SNPs with global MAF $\geq 5\%$ on these chromosomes, respectively. Manhattan plots of sdMAF p-values (S15–S17 Figs), as well as the histograms and QQ plots (S18 Fig), show that there are very few SNPs on these autosomes with genome-wide significant sdMAF. Specifically, only 6, 1 and 0 SNPs on chromosomes 1, 7 and 22 had sdMAF p-values <5e-8, respectively. These numbers are significantly lower than that found for the X chromosome.

To obtain better insight into the nature and source the sdMAF on the autosomes, we then further examined six SNPs, two from each of the three autosomes with the smallest sdMAF p-values (S1 Data). For example, on chromosome 1 rs10803097 (GRCh37 position = 243050350; sdMAF p-value = 2.15e-25) has higher MAFs in males than females in all five super-populations, but the population-specific sdMAFs are only genome-wide significant in EAS and AFR (S1 Data) while sdMAF p-value = 0.12 in SAS, suggesting heterogeneity in sdMAF between the super-populations at this autosomal SNP. The HWE testing are genome-wide significant in males in all super-populations (except SAS) with excess of heterozygous males, but not in females. A BLAST [28] search of a 100 nucleotide sequence flanking this SNP identified perfect match to sequence on the NPR region of the Y chromosome (GRCh38 position:11786038), with the Y chromosome having the alternate allele at the SNP, suggesting that it is a Paralogous Sequence Variant [29].

In contrast, on chromosome 7 rs78984847 (GRCh37 position = 72053830; sdMAF p-value = 3.5e-18) has higher MAFs in females than males in all five super-populations. The population-stratified sdMAF p-values are consistently small, 7.66e-5, 3.06e-4, 1.76e-5, 7.52e-3, and 1.79e-6 respectively in EAS, EUR, AFR, AMR and SAS, but not genome-wide significant as a result of reduced sample sizes. In addition, HWE p-values are much smaller in females than males for all five super-populations, with excess of heterozygous females; the HWE p-values in females are <5e-8 in EUR, AFR and SAS, and 2.92e-7 in EAS and 1.48e-5 in SAS (S1 Data). There is also evidence for departure from HWE in males; the HWE p-values in males are 2.02e-2, 1.67e-3, 1.21e-4, 2.57e-4 and 1.57e-2 respectively in EAS, EUR, AFR, AMR, and SAS, with excess of CC males. BLAST of a 100 nucleotide sequence centred on this SNP identified multiple close matches to other chromosomes, including the X chromosome.

## Comparison for specific SNPs between the phase 3 and high coverage sequence data of the X chromosome

sdMAFs and sex-specific genotype agreements between the two phases were attempted for 130 SNPs, selected based on the smallest sdMAF p-values in the phase 3 data (GRCh37) from the four regions (NPR, PAR1, PAR2, and PAR3). We required success of liftover onto GRCh38, that they were biallelic in both datasets, and with no genotype missingness. The liftover failure rates differed by region: NPR = 92%, PAR1 = 0%, PAR2 = 55%, PAR3 = 82%. Using all criteria left 33 SNPs: 4 in NPR, 10 in PAR1, 10 in PAR2, and 9 in PAR3 (S2 Note with SNPs ordered by the GRCh37 positions).

For the 10 SNPs in PAR1, the high coverage data did not resolve the genome-wide significant sdMAF observed in the phase 3 data (Tables A-AD (pages 2–11) in S2 Note). In addition,

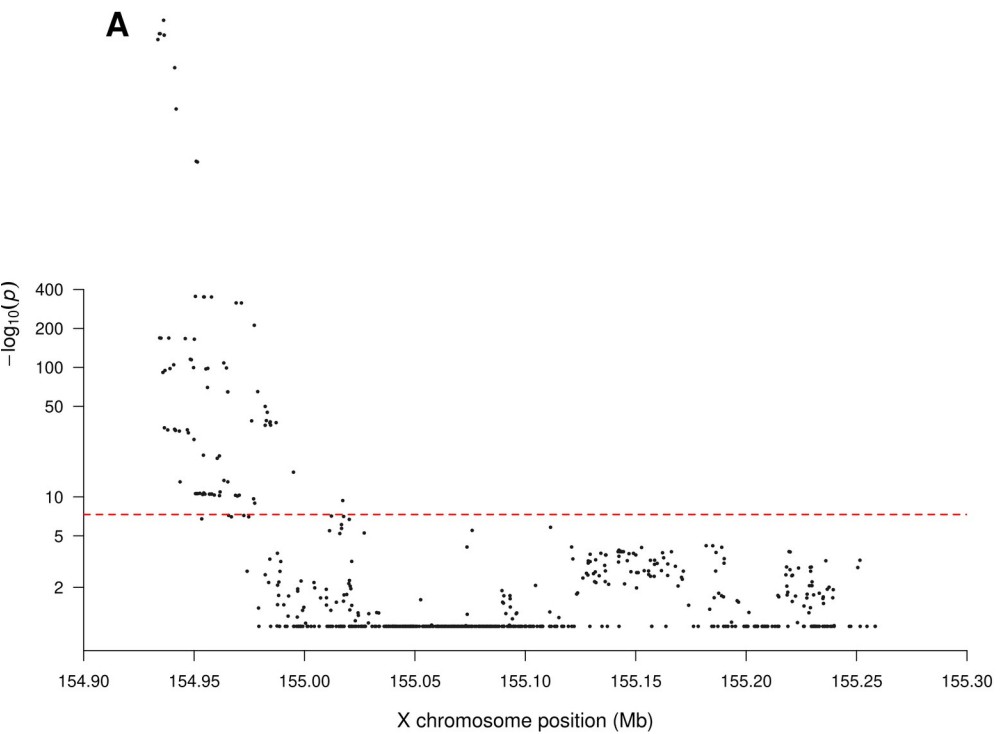

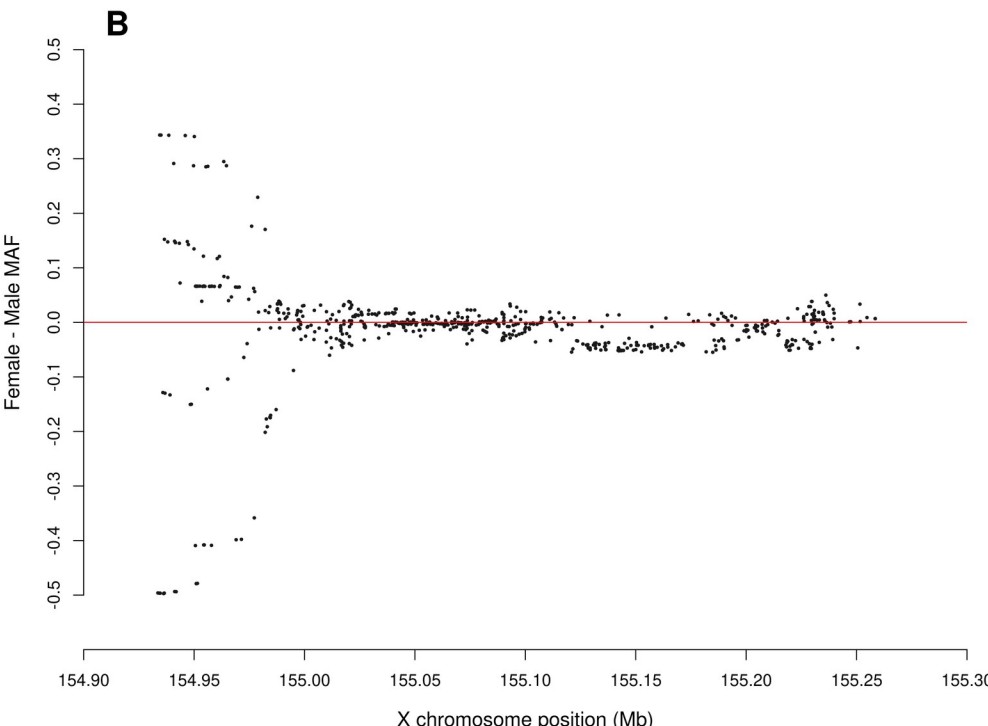

**Fig 5. Zoomed-in plot for testing for sex difference in MAF across PAR2 of the X chromosome from the 1000 Genomes Project phase 3 data on GRCh37.** A: sdMAF p-values for bi-allelic SNPs with global MAF ≥5% presumed to be of high quality. Y-axis is −log10(sdMAF p-values) and p-values >0.1 are plotted as 0.1 (1 on −log10 scale) for better visualization. The dashed red line represents 5e-8 (7.3 on the −log10 scale). B: Female—Male sdMAF for the

same SNPs in part A, clearly showing PAR2 SNPs with significant sdMAF tend to cluster at the NPR-PAR2 boundary around 88.5 Mb.

they showed good genotype agreement with the phase 3 data; similar results were observed for 10 SNPs in PAR2 (Tables BR-CU (pages 25–34) in S2 Note). For the 4 SNPs in NPR (Tables AE-AM (pages 12–14) and Tables BO-BQ, and (page 24) in S2 Note), sdMAFs are no longer genome-wide significant in the high coverage data, suggesting genotyping error in phase 3. For the 9 SNPs in PAR3 (Tables AN-BN (pages 15–23) in S2 Note), two persisted with genome-wide sdMAF in the high coverage data, located at the centromeric boundary between PAR3 and NPR, while the remaining seven in PAR3 were no longer significant.

### High coverage 1000 Genomes *X chromosome-wide sdMAF*

Results of X chromosome-wide sdMAF analysis of the high coverage data, without the liftover restriction, are reported in S3 Table and S19–S25 Figs. S19 Fig shows the analytical pipeline of selecting bi-allelic SNPs with population-pooled MAF ≥5% using the high coverage data, and S3 Table contains the counts of variants by region, MAF threshold and those excluded.

The Manhattan plot of sdMAF p-values in S20 Fig shows that, as compared to Fig 2 for the phase 3 data, the prevalence of genome-wide significant sdMAF is reduced in the high coverage data for NPR and PAR3, about a 10-fold deduction: 0.11% of SNPs in NPR and 0.07% in PAR3 have sdMAF p-values <5e-8 in the high coverage data, as compared to 0.83% in NPR and 0.85% in PAR3 in phase 3. This suggests that genotyping error is a contributing factor to some of the significant sdMAF observed in the phase 3 data. The causes of the remaining sdMAF in NPR and PAR3 in the high coverage data require further examination.

The sdMAF results for PAR1 (Figs 4 and S21) and PAR2 (Figs 5 and S22) are practically the same between the two phases: 0.30% of SNPs in PAR1 and 12.2% in PAR2 have sdMAF p-values <5e-8 in the high coverage data, as compared to 0.29% in PAR1 and 13.1% in PAR2 in phase 3. The sdMAF SNPs in the high coverage data still occur at the NPR-PAR1 and NPR-PAR2 boundaries, which are evident from the zoomed-in Manhattan plots (S21 and S22 Figs).

SNPs with significant sdMAF also remain at the centromeric NPR-PAR3 boundary (S23 Fig). The persistent presence of small sdMAF p-values, including in the NPR region, is also evident from the QQ plots (S24 Fig) and histograms of p values (S25 Fig). This strongly suggests that sex-linkage is a major driver for sdMAF at both PAR1 and PAR2, and at the centromeric boundary of PAR3. Association studies of the X chromosome thus must consider sdMAF.

### X chromosome sdMAF analysis of high-coverage whole genome sequence data from gnomAD v3.1.2

To examine consistency of sdMAF in other high coverage whole genome sequence data, we used the genotype and allele counts from gnomAD separately from the two largest populations: the non-Finnish Europeans and Africans/African Americans to examine sdMAF on the X chromosome.

To be consistent with the 1000 Genomes Project analysis, the primary analyses were restricted to SNPs with MAF>5% and where the majority of samples had genotypes. This resulted in 53,002 SNPs in non-Finnish Europeans and 97,438 in African/African Americans (S26 and S27 Figs, respectively). These analyses confirm the existence of many regions with sdMAF that we documented in the 1000 Genomes Project data, including the PAR1-NPR

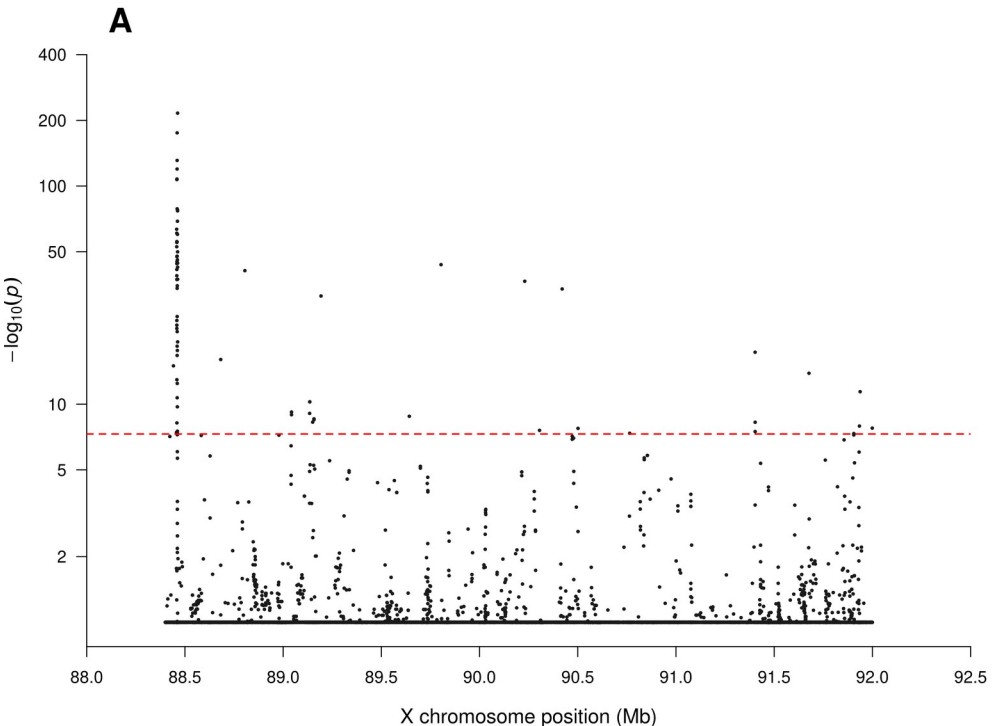

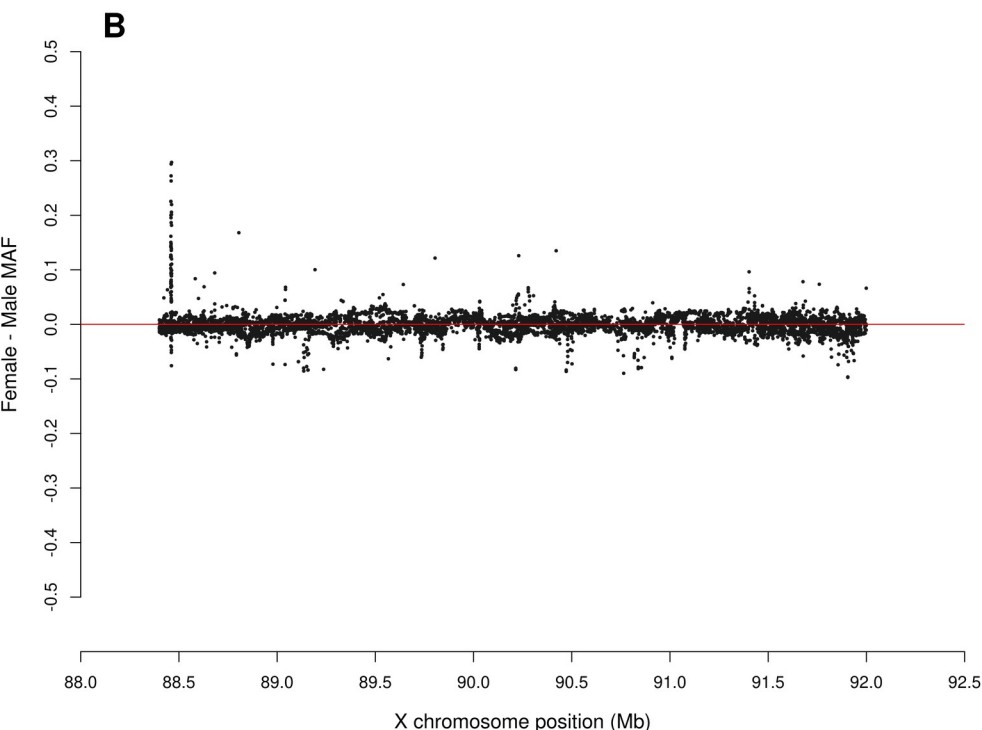

**Fig 6. Zoomed-in plot for testing for sex difference in MAF across PAR3 of the X chromosome from the 1000 Genomes Project phase 3 data on GRCh37.** A: sdMAF p-values for bi-allelic SNPs with global MAF ≥5% presumed to be of high quality. Y-axis is −log10(sdMAF p-values) and p-values >0.1 are plotted as 0.1 (1 on −log10 scale) for better visualization. The dashed red line represents 5e-8 (7.3 on the −log10 scale). B: Female—Male sdMAF for the

same SNPs in part A, clearly showing PAR3 SNPs with significant sdMAF tend to cluster at one of the NPR-PAR3 boundaries around 88.5 Mb.

boundary, the PAR2-NPR boundary and the centromeric boundary of PAR3 (Figs 8 and 9, respectively). Online locuszoom plot [30] for the non-Finnish European data is available: https://my.locuszoom.org/gwas/717341/?token=b784386eb4574ef7ba46c117ed711ccf

To illustrate the sdMAF results for specific SNPs, we selected eight SNPs with the most significant sdMAF from the gnomAD non-Finnish European and African/African American populations, one each from the four regions (Table 1). There is remarkably consistency of sdMAF direction and magnitude across these two populations. Of note, the SNP genotype call

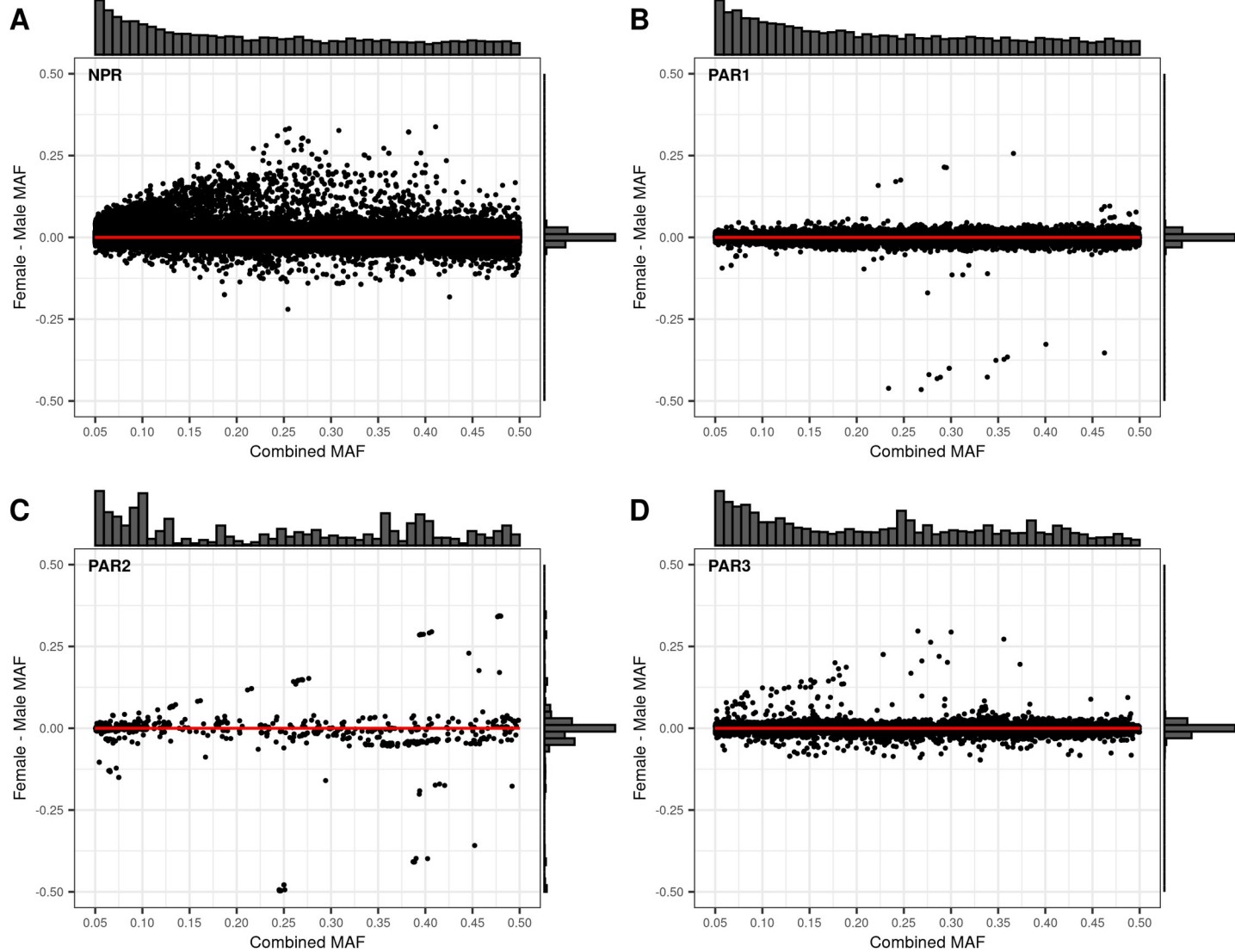

**Fig 7. Bland-Altman plots comparing the Female—Male sex difference in MAF to the sex-combined MAF across the X chromosome from the 1000 Genomes Project phase 3 data on GRCh37.** Regions are plotted separately A: NPR; B: PAR1, C: PAR2; D: PAR3. For each of the four regions, the histogram at the top of the Bland-Altman plot shows the distribution of the sex-combined MAF for bi-allelic SNPs with global MAF $\geq$5% presumed to be of high quality. The histogram to the right of the plot shows the distribution of the Female—Male sdMAF.

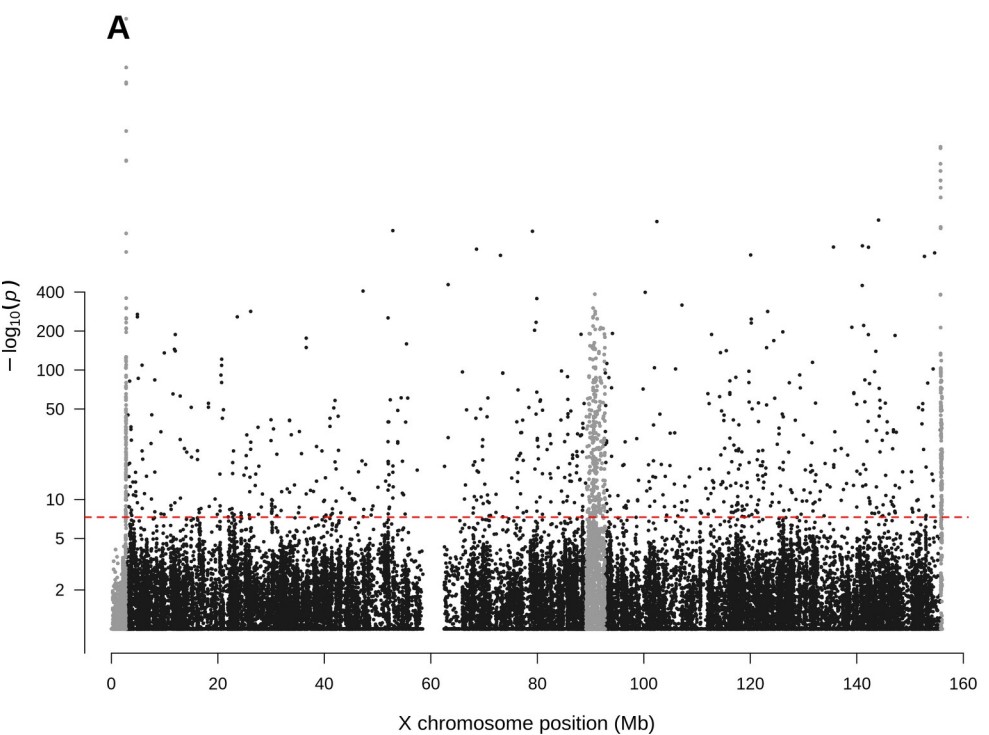

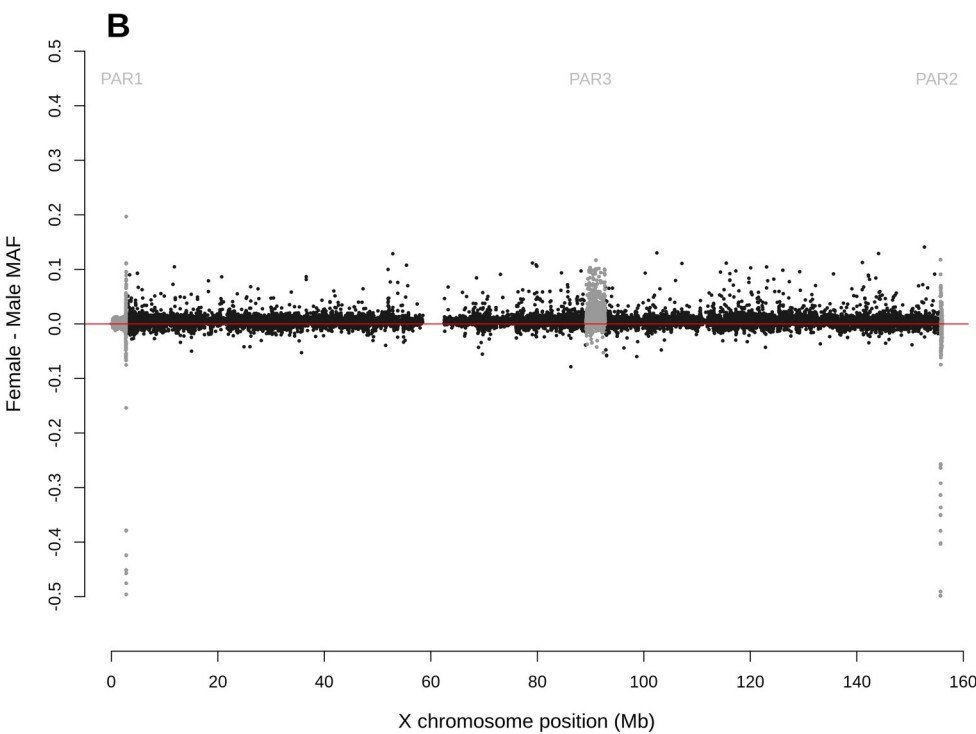

**Fig 8. Manhattan plot for testing for sex difference in MAF across the X chromosome of the Non-Finnish European population from the gnomAD v3.1.2 data on GRCh38.** A: sdMAF p-values for bi-allelic SNPs with MAF $\geq$5% and the total number of alleles >50,000 in the Non-Finnish European population, presumed to be of high quality. SNPs in the PAR1, PAR2 and PAR3 regions are plotted in grey, with PAR3 located around 90 Mb. Y-axis is -log10 (sdMAF p-values) and p-values >0.1 are plotted as 0.1 (1 on -log10 scale) for better visualization. The dashed red line represents 5e-8 (7.3 on the -log10 scale). B: Female—Male sdMAF for the same SNPs in part A.

rate in males are generally lower than females (paired non-parametric test p value = 0.0026) for these SNPs, which may provide insight into some mechanisms resulting in sdMAF.

## Rarer variants in gnomAD v3.1.2 non-Finnish Europeans

To determine whether sdMAF also is present for rarer variants, we used the non-Finnish Europeans from gnomAD v3.1.2. to examine 155,828 X chromosomal variants with MAF 0.1% to 5%. These would typically have a minimum minor allele count of 50. Significant sdMAF was observed at the PAR1 and PAR2 boundaries with NPR, as well as those in PAR3 (S28 Fig). In addition, there were other regions including around 142 Mb, where there were clusters of rarer SNPs with significant sdMAF where female MAFs were greater than male. This later region was not observed strongly in the variants with MAF>5% in the same data (Fig 8). Online locuszoom of these results is available: https://my.locuszoom.org/gwas/473034/? token=78b9004c71f04892adf5f514be014044

## Discussion

Our initial sdMAF analysis focused on the 1000 Genomes Project phase 3 data since it has been examined extensively for association analysis, and is one of the most commonly used imputation panels for GWAS [31]. With the recent release of the high coverage data, we first compared results between the two phases for specific SNPs. In addition, we also performed a separate X chromosome-wide analysis of the 1000 Genomes Project high coverage data aligned to GRCh38 as well as mostly high-coverage whole genome sequence aligned to GRCh38 from two populations from gnomAD.

In the 1000 Genomes Project phase 3 data, focused on eight selected SNPs, we showed that the sdMAF was robust to population stratification either coming from super-population or population levels. Yet, we cannot exclude the possibility that there could be additional SNPs with sdMAF that we have missed. Similar concerns relate to the analysis of the gnomAD data, where our analysis was performed separately for the two major super-populations (likely defined by autosomal population structure), the results could be confounded by population structure and/or admixture within them. However, S10 Fig shows that sdMAF estimates are consistent between the five super populations across the whole X chromosome in the 1000 Genomes phase 3 data. Additionally, Fig 3 shows that the sdMAF estimates are extremely consistent between the 26 populations for the eight selected SNPs with the smallest sdMAF p value.

We identified two likely sources of sdMAF: genotyping error and sex-linkage. Genotyping error (which may in part be due to differences between GRCh37 and 38) accounts for many NPR and PAR3 sdMAF in phase 3 since they were mostly resolved in the high coverage data. However, sdMAF for some NPR and PAR3 SNPs remain in the high coverage data and are also present in gnomAD. Thus, despite the recent advances in how to analyze NPR SNPs [6], our findings here show that robust X chromosomal association methods must consider sdMAF caused by genotyping error and/or sex-specific selection. For example, for sex-dimorphic traits, others have suggested sex-specific analysis genome-wide [32]. Updated reference

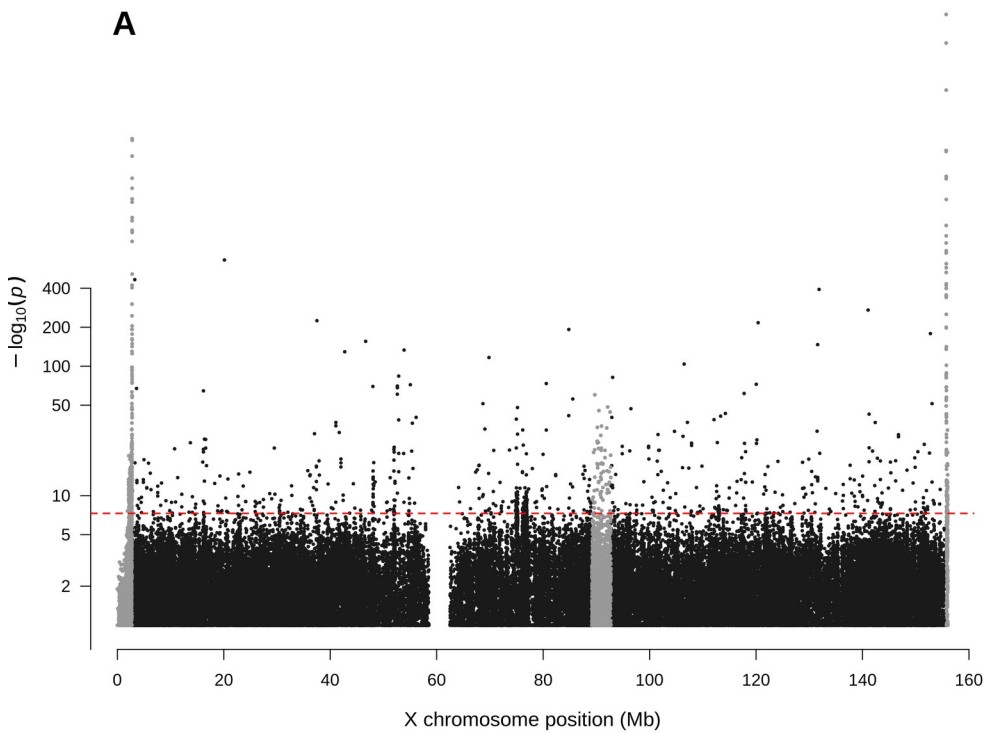

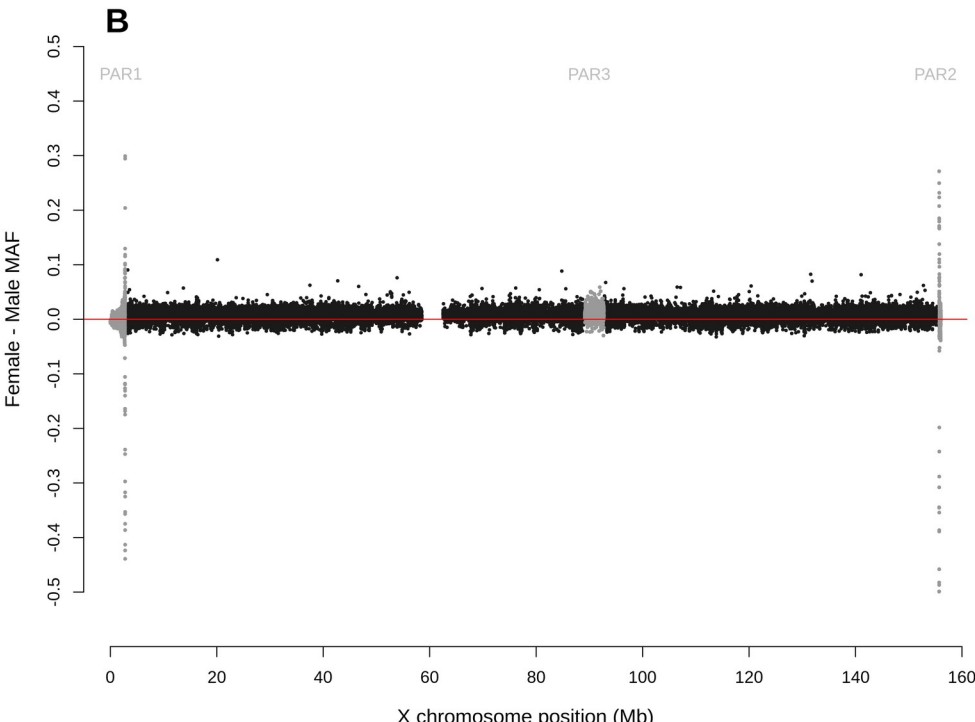

**Fig 9. Manhattan plot for testing for sex difference in MAF across the X chromosome of the African/African American population from the gnomAD v3.1.2 data on GRCh38.** A: sdMAF p-values for bi-allelic SNPs with MAF ≥5% and the total number of alleles >30,000 in the African/African American population, presumed to be of high quality. SNPs in the PAR1, PAR2 and PAR3 regions are plotted in grey, with PAR3 located around 90 Mb. Y-axis is

-log10(sdMAF p-values) and p-values >0.1 are plotted as 0.1 (1 on -log10 scale) for better visualization. The dashed red line represents 5e-8 (7.3 on the -log10 scale). B: Female—Male sdMAF for the same SNPs in part A.

sequence along with alternate haplotypes for regions of the X and Y chromosomes may improve the mapping of sequence reads and genotype quality [33,34].

In contrast, the impact of sex-linkage in PAR1 and PAR2 results in most sdMAF identified in phase 3 persisting in the high coverage data. Sex-linkage at PARs has previously been discussed for linkage analyses using affected sibpairs [18], but it has not been examined in the context of association studies [16,17]. Multiple authors have stated that association methods

**Table 1. SNPs with most significant sdMAF from gnomAD v 3.1.2 genome in non-Finnish European and African/African Americans**

| X chr Region | SNP (GRCh38) | gnomAD Population | Sex | Call rate (%) | RR | RA | AA | Minor Allele | sdMAF | sdMAF.p | HWD.delta | HWE.p |
|---|---|---|---|---|---|---|---|---|---|---|---|---|
| PAR1 | X-2779827-G-A | NFE | F | 99.72 | 19436 | 194 | 1 | A | | | 0.000 | 0.46 |
| PAR1 | X-2779827-G-A | NFE | M | 99.50 | 626 | 13577 | 67 | A | -0.475 | <1E-300 | -0.226 | <1E-300 |
| PAR1 | X-2779827-G-A | AFR | F | 99.16 | 6772 | 3707 | 514 | A | | | 0.000 | 0.82 |
| PAR1 | X-2779827-G-A | AFR | M | 99.08 | 248 | 7346 | 1975 | A | -0.375 | <1E-300 | -0.142 | <1E-300 |
| PAR1 | X-2780882-G-A | NFE | F | 99.85 | 19567 | 89 | 0 | A | | | 0.000 | 0.75 |
| PAR1 | X-2780882-G-A | NFE | M | 99.64 | 91 | 14150 | 39 | A | -0.496 | <1E-300 | -0.245 | <1E-300 |
| PAR1 | X-2780882-G-A | AFR | F | 99.51 | 8781 | 2118 | 133 | A | | | 0.000 | 0.68 |
| PAR1 | X-2780882-G-A | AFR | M | 99.64 | 137 | 8442 | 1044 | A | -0.439 | <1E-300 | -0.191 | <1E-300 |
| NPR | X-52861869-T-C | NFE | F | 98.54 | 14176 | 5222 | 1 | C | | | -0.018 | 1.45E-103 |
| NPR | X-52861869-T-C | NFE | M | 85.50 | 12189 | NA | 73 | C | 0.129 | <1E-300 | NA | NA |
| NPR | X-52861869-T-C | AFR | F | 99.36 | 9642 | 1368 | 5 | C | | | -0.003 | 6.00E-10 |
| NPR | X-52861869-T-C | AFR | M | 90.36 | 8552 | NA | 175 | C | 0.042 | 1.43E-84 | NA | NA |
| PAR3 | X-89688092-A-G | NFE | F | 99.48 | 13994 | 5106 | 483 | G | | | 0.001 | 0.50 |
| PAR3 | X-89688092-A-G | NFE | M | 89.78 | 11538 | NA | 1338 | G | 0.051 | 1.37E-55 | NA | NA |
| PAR3 | X-89688092-A-G | AFR | F | 99.56 | 9694 | 1300 | 43 | G | | | 0.000 | 0.93 |
| PAR3 | X-89688092-A-G | AFR | M | 84.52 | 7865 | NA | 298 | G | 0.026 | 2.44E-23 | NA | NA |
| PAR3 | X-90958827-T-C | NFE | F | 99.38 | 15016 | 4218 | 330 | C | | | 0.001 | 0.09 |
| PAR3 | X-90958827-T-C | NFE | M | 91.08 | 12039 | NA | 1025 | C | 0.046 | 1.66E-57 | NA | NA |
| PAR3 | X-90958827-T-C | AFR | F | 99.31 | 9611 | 1353 | 45 | C | | | 0.000 | 0.72 |
| PAR3 | X-90958827-T-C | AFR | M | 85.60 | 7990 | NA | 277 | C | 0.032 | 3.17E-35 | NA | NA |
| NPR | X-141051591-A-G | NFE | F | 99.31 | 14846 | 4700 | 5 | G | | | -0.014 | 6.07E-79 |
| NPR | X-141051591-A-G | NFE | M | 79.08 | 11253 | NA | 89 | G | 0.113 | <1E-300 | NA | NA |
| NPR | X-141051591-A-G | AFR | F | 98.57 | 8826 | 2095 | 7 | G | | | -0.009 | 2.57E-25 |
| NPR | X-141051591-A-G | AFR | M | 84.68 | 8057 | NA | 121 | G | 0.082 | <1E-300 | NA | NA |
| PAR2 | X-155706522-G-A | NFE | F | 98.23 | 19321 | 16 | 0 | A | | | 0.000 | 0.95 |
| PAR2 | X-155706522-G-A | NFE | M | 97.92 | 30 | 14012 | 2 | A | -0.499 | <1E-300 | -0.249 | <1E-300 |
| PAR2 | X-155706522-G-A | AFR | F | 98.71 | 10933 | 10 | 0 | A | | | 0.000 | 0.96 |
| PAR2 | X-155706522-G-A | AFR | M | 99.26 | 11 | 9573 | 3 | A | -0.499 | <1E-300 | -0.249 | <1E-300 |
| PAR2 | X-155712209-T-G | NFE | F | 99.63 | 19585 | 28 | 0 | G | | | 0.000 | 0.92 |
| PAR2 | X-155712209-T-G | NFE | M | 99.12 | 29 | 14186 | 2 | G | -0.498 | <1E-300 | -0.249 | <1E-300 |
| PAR2 | X-155712209-T-G | AFR | F | 99.58 | 10520 | 516 | 3 | G | | | 0.000 | 0.19 |
| PAR2 | X-155712209-T-G | AFR | M | 99.78 | 14 | 9404 | 218 | G | -0.487 | <1E-300 | -0.238 | <1E-300 |

SNP name is from gnomAD which incorporates chromosome-nucleotide position and reference-alternate alleles; Population NFE = non-Finnish European; AFR = African/African American. Sex: F = female, M = male; call rate is proportion of individuals with non-missing genotypes; RR = reference homozygote/hemizygote genotype count; RA = heterozygote genotype count; AA = alternate allele homozygote/hemizygote genotype count. sdMAF is female-male allele frequency difference, along with p value. HWD delta is the Hardy-Weinberg equilibrium delta, along with p value, NA = not applicable.

routinely used for autosomes can be applied to the PARs [3–5]. However, our results indicate that association analyses at the PAR-NPR boundaries (and at the centromeric boundary of PAR3) should consider sdMAF caused by sex-linkage. The optimal region-specific solution is an open research question.

We searched the NHGRI-EBI GWAS catalog [35] and identified multiple signals at the PAR boundaries. These include loci: for BMI, multiple lipids, red cell traits, non-syndromic metopic craniosynostosis at the PAR2 boundary [36–39]; ANCA-associated vasculitis, Alzheimer's disease, non-syndromic metopic craniosynostosis, susceptibility to TB, 3-hydroxy-1-methylpropylmercapturic acid levels and adenosine diphosphate at PAR3 [36,40–45]; age of onset of myopia, mean corpuscular volume, eosinophil count, and asthma [43][39,46,47] close to PAR1/NPR boundary. Apart from two studies [40,45], none of others provided sex-specific results, making it difficult to determine the effect of sdMAF on these associations.

The PAR1-NPR boundary is in intron 3 of XG. The genetic basis of the Xg$^a$ blood group has recently been studied [48–50] with groups independently identifying rs311103 in PAR1 as the potential causal variant. Other work has also suggested that a large deletion that spans PAR1-NPR could also be a separate causal variant for Xg$^a$ [51]. Polymorphic duplications of the Y chromosome neighbouring PAR1, as well as a deletion of the overlapping region on the X chromosome have been characterized [52]. Since these CNVs are relatively rare, e.g. as described in gnomAD v 2.1 (Web Resources) [53], they are unlikely to result in the major sdMAF seen at the PAR1-NPR boundary.

A small proportion (3%) of sdMAF in phase 3 were in the 19 X/Y homologs. Further examination of allele frequencies of the Y chromosome is interesting but beyond the scope of this work.

It is unlikely that strand flips are a major cause of the difference in genotype by sex, since variant calling was performed blind to sex and typically joint-called, especially for the high coverage 1000 Genomes Project data [54] and the gnomAD data [55]. Further, inspection of Integrative Genomics Viewer (IGV) plots in gnomAD v3.1.2 shows that there are few discrepancies of sequencing reads around these SNPs, making it unlikely that they are strand flips. Sex-specific quality metrics for variants on the X chromosome could help identify variants with technical interference.

Of note, for many variants with sdMAF, there was also deviation from Hardy-Weinberg equilibrium [27] in each of the super-populations in females (NPR and PAR3) and males (PAR1 and PAR2) (S1 Data). HWD testing of variants in the 1000 Genomes phase 3 data has been examined separately in the JPT and YRI populations [56,57]. SNPs with missing rate <5% and possessing an rs identifier were used. The earlier work [56] found lower rates of deviation from HWE on the X chromosome (after exclusion of PAR1 and PAR2) than on any of the autosomes, but the sample size for the X chromosome is ~3/4 of that for the autosomes. Results from our sdMAF analysis also calls for new X chromosome-aware HWD methods that consider sdMAF.

Earlier work has examined HWD of X chromosomal variants (59), using a 2 degrees of freedom (df) Pearson Chisq-based test that jointly analyzes both females and males. The test includes, in addition to the female data, the deviation of male genotype counts from the expected, based on *pooled* allele frequency estimate using both male and female data. However, it has been shown that this 2 df HWD test is equivalent to testing for HWD in females alone and simultaneously testing for sdMAF between the sexes [58]. Because of this confounding between HWD and sdMAF, for variants in the PAR1 and PAR2 regions, we performed the HWD analysis stratified by the sex. And for NPR and PAR3 variants, we performed the HWD analysis in females only; the males have two hemizygous genotypes, leaving no df to perform the HWD test after using one df to estimate the MAF in males.

In addition to the 2 df Chisq test, the work of [57] also proposed and recommended an Exact test, jointly analyzing both females and males. Although the Exact test does not specify the degrees of freedom, our preliminary work suggests that it is similar to the 2 df Chisq test in spirit, because the Exact test includes males and is derived conditional on the "number of A alleles (nA) and the observed male and female frequencies." This could lead to increased type I error HWD test in the presence of true sdMAF. For example, consider a SNP from the NPR region with significant sdMAF but no deviation from HWE in females: female MAF = 0.2 with genotype counts of (640, 320, 40) and male MAF = 0.5 with genotype counts of (500, NA, 500). In that case, the sex-combined EXACT test p = 6.41E-61 (using the R program HardyWein-berg v1.7.4 [59]), but this highly significant Exact HWD testing result is due to the significant sdMAF alone. When the HWD analysis is performed using only females and in the presence of small sample, the Exact test can be advantageous [56,57].

Beyond the X chromosome, joint and separate analyses of sdMAF and HWD have also been performed for autosomal variants [60]. Interestingly, an application to autosomal SNPs using the 104 individuals from the JPT sample of the 1000 Genomes Project showed that, in the presence of sdMAF, true HWD may go unnoticed if applying the standard HWD testing using both males and females.

How to solve the visualization of Manhattan/Miami plots for the X chromosome stratified by sex, while also highlighting the four regions, is a worthy challenge for bioinformaticians. Additionally, as both the association and sdMAF analyses differ between regions, it is crucial to assign variants at the NPR-PAR boundaries to the correct region.

Since the 2013 call for X chromosome-inclusive GWAS [1] several methods have been developed, all focused on robustifying the association analysis against the well-known phenomenon of X-inactivation uncertainty [2,3,5–12]. However, the sex differences in minor allele frequencies phenomenon documented here could affect the validity of existing X-inactivation-aware association methods against spurious sdMAF due to, for example, genotyping error. As sex is typically included in X chromosomal association analysis, it is reasonable to assume that sdMAF is accounted for through the inclusion of sex as covariate. However, further research is needed particularly for studies of traits displaying significant sexual dimorphism. For those traits, there could be a sex-ratio difference between the cases and controls, and this combined with sdMAF could have implications for association studies of the X chromosome. Finally, sdMAF at the NPR-PAR boundaries is likely a biological phenomenon. Thus, how to leverage true sdMAF to increase association power is also an open research question.

## Materials and methods

### The X chromosome phase 3 data of the 1000 Genomes Project

The 1000 Genomes Project generated an integrated call set of variants for phase 3 (release 5) data based on four data types: Illumina 2.5M genotyping array, Affymetrix SNP6.0, high-coverage whole exome sequence (WES), and low coverage whole genome sequence (WGS) [61]. Because only a minority of variant positions are covered by the first three, >93% of the data come from the WGS, which is generally of low quality due to its low-coverage nature (average depth of 7.4 on the autosomes).

The phase 3 data of the 1000 Genomes Project identified 3,468,093 variants on the X chromosome [61], with an average depth of 6.2. Assuming equal proportions of males and females, the sex-specific average depth is 7.0 for females and 3.5 for males.

Focusing on bi-allelic common SNPs presumed to be of high quality, we first removed variants that had >2 alleles, as well as indels. We then analysed all the bi-allelic SNPs from the

whole of the X chromosome, including the pseudo-autosomal region 1, PAR2 and PAR3 [19], in addition to the non-pseudo-autosomal region. X chromosomal locations for these four regions, NPR, PAR1, PAR2, and PAR3, were obtained from The Genome Reference Consortium, based on GRCh37.

SNPs were included based on global minor allele frequency ≥5%, because they have been shown to have the highest quality in this data [61]. We defined the minor allele based on the overall sample of the 1000 Genomes Project, with males providing a single allele count in the non-PAR regions for the MAF calculation. The definition of a minor allele has no necessary relationship to the reference allele in GRCh37, and the minor allele defined in the sex- and population-pooled sample may not be minor in sex- or population-stratified samples. The counts of variants by region, MAF threshold and those excluded are provided in S1 Table; the analytic pipeline and flow-chart of the analyses are provided in Fig 1; Supplementary codes for all the analytical steps are also provided (Web Resource).

### Testing for X chromosomal variants with sex difference in MAF (sdMAF)

Sex difference in MAF is an indication of either potential genotyping error or a biological phenomenon particularly at the NPR-PAR boundaries. Although sdMAF analyses could be performed separately for each of the five super-populations, or each of the 26 populations, this is not a powerful approach. As the sex ratio is similar across the populations (S1 Fig), we tested sdMAF using the whole sample of the 1000 Genomes Project. This more powerful approach can detect sex difference in MAF if it is present in any of the sub-samples. If there are no sdMAFs in any of the sub-samples, the test remains valid (i.e. accurate control of false positives) even if the MAFs differ drastically between the sub-populations. This is because the sdMAF test detects the difference in MAF *between females and males*, not the difference in MAF between populations.

**1.1. sdMAF test.** For each bi-allelic SNP in the NPR and PAR3 regions of the X chromosome, notations in S4A Table denote sex-stratified genotype counts, where, without loss of generality, allele *A* represents the minor allele defined in the sex-pooled whole sample.

To identify variants in the NPR and PAR3 regions with sex difference in MAF, we used the following conservative test statistic,

$$T_X = \frac{\left(\widehat{p}_f - \widehat{p}_{m,X}\right)^2}{\frac{1}{2f}\left[\widehat{p}_f\left(1 - \widehat{p}_f\right) + \widehat{\delta}_f\right] + \frac{1}{m}\left[\widehat{p}_{m,X}\left(1 - \widehat{p}_{m,X}\right)\right]} \overset{H_0}{\sim} \chi_1^2, \tag{1}$$

where the numerator, $\left(\widehat{p}_f - \widehat{p}_{m,X}\right)$, contrasts the frequency estimates of allele *A* between the female and male groups, and the denominator is the estimate of the variance of $\left(\widehat{p}_f - \widehat{p}_{m,X}\right)$ while allowing for Hardy-Weinberg disequilibrium (HWD) in females via $\widehat{\delta}_f$. Specifically,

$$\widehat{p}_{m,X} = \frac{m_2}{m}, \quad \widehat{\text{Var}}\left(\widehat{p}_{m,X}\right) = \frac{\widehat{p}_{m,X}\left(1 - \widehat{p}_{m,X}\right)}{m}, \tag{2}$$

$$\widehat{p}_f = \frac{2f_2 + f_1}{2f}, \quad \widehat{\text{Var}}\left(\widehat{p}_f\right) = \frac{\widehat{p}_f\left(1 - \widehat{p}_f\right) + \widehat{\delta}_f}{2f}, \tag{3}$$

where

$$\widehat{\delta}_f = \widehat{p}_f(AA) - \widehat{p}_f^2 = \frac{f_2}{f} - \left(\frac{2f_2 + f_1}{2f}\right)^2 \tag{4}$$

is the estimate of HWD present in females [62]. Under the null of no sdMAF, $H_0$, the test statistic $T_X$ is asymptotically $\chi_1^2$ distributed, as it is a straightforward application of the classic two-sample comparison with the consideration of HWD. Note that the $H_0$ of interest here refers to no sex difference in MAF, while allowing for population difference in MAF.

When applied to a sample consisting of individuals from multiple populations (e.g. the whole sample of the 1000 Genomes Project), the sdMAF test based on $T_X$ is conservative because the denominator in Eq (1) contains a population-pooled HWD estimate. Note that even if each of the five super-populations or 26 populations is in Hardy-Weinberg equilibrium (HWE), the combined population may not be in HWE unless the MAFs are the same across all populations [63]. In addition, we show in the S1 Note that the bias factor for the population-pooled HWD estimate is always greater or equal to zero, resulting in $T_X$ being (slightly) conservative in testing for sdMAF.

We note that using a conservative sdMAF test is not an issue for the purpose of this study, because for SNPs declared to have significant sdMAF we are then more confident about the conclusion. Likewise, we used the genome-wide significance level of p-value <5e-8 [64] to declare sdMAF significance, which is conservative for our X chromosome-focused analysis.

S4B Table shows the genotype counts for each bi-allelic SNP in the PAR1 and PAR2 regions, for which a male has three genotypes as for an autosomal SNP. In that case, we used the following test statistic to test for sdMAF,

$$T_A = \frac{\left(\widehat{p}_f - \widehat{p}_m\right)^2}{\frac{1}{2f}\left[\widehat{p}_f\left(1 - \widehat{p}_f\right) + \widehat{\delta}_f\right] + \frac{1}{2m}\left[\widehat{p}_m(1 - \widehat{p}_m) + \widehat{\delta}_m\right]} \overset{H_0}{\sim} \chi_1^2, \tag{5}$$

where all notations with subscripts $_f$ are the same as in Eqs (3) and (4), while for males,

$$\widehat{p}_m = \frac{2m_2 + m_1}{2m}, \ \widehat{\mathrm{Var}}(\widehat{p}_m) = \frac{\widehat{p}_m(1 - \widehat{p}_m) + \widehat{\delta}_m}{2m}, \tag{6}$$

where

$$\widehat{\delta}_m = \widehat{p}_m(AA) - \widehat{p}_m^2 = \frac{m_2}{m} - \left(\frac{2m_2 + m_1}{2m}\right)^2. \tag{7}$$

We note that, in finite sample, the Wald's test shown in Eq (5) is more conservative than the Score test, where the sex-stratified MAF and HWD estimates in the denominator of [5] would be replaced with sex-pooled estimates; asymptotically the two tests are equivalent.

## Super-population specific analysis and meta-analysis

As a second complimentary analysis we also performed super-population specific analysis, as well as meta-analysis of the super-populations. The meta-analysis test statistic is sample size based [65], which takes the weighted average of Z-scores with the square root of sample size as weights. We limited the analysis to bi-allelic SNPs with global MAF ≥5% in the combined sample, and also polymorphic in each of the five superpopulations to have a valid and comparable meta-analysis statistics across SNPs. Due to this constraint, there is a reduction in the number of SNPs analyzed in meta-analysis compared to the primary analysis.

In each super-population, a SNP's alleles can be sex-specific, for example where one allele is present only in females, and the other is present only in males (MAF would be > 5%). In that case, the sdMAF test statistic would be infinity due to the variance being zero in the super-population sdMAF analysis, which leads to meta-analysis p-value being zero. In those cases, the p-values are plotted as 47,982.36 on the–log10 scale, the smallest p-value observed in the mega-analysis.

## In-depth analyses of eight X chromosomal SNPs with genome-wide significant sdMAF in the phase 3 data

To better understand the patterns of sdMAF, we selected eight SNPs for additional analyses, two from each of the four regions (NPR and PAR1-3) with the smallest sdMAF p-values in phase 3 data of the 1000 Genomes Project. For each SNP, we calculated population-specific and sex-stratified allele frequency estimates; for a NPR or PAR3 SNP, each male only contributed a single allele count to the allele frequency calculation.

For the four SNPs in the NPR and PAR3 regions, we also performed population-stratified female-only HWE testing for each of the five super-populations, using the standard autosomal method as only females were analyzed here. That is, we used $\widehat{\delta}_f$ in Eq (4) to estimate HWD and the following Pearson's chi-square test statistic to test for HWD in females,

$$T_{\text{HWE, female}} = \frac{\left(f_2 - f\left(1 - \widehat{p}_f\right)^2\right)^2}{f\left(1 - \widehat{p}_f\right)^2} + \frac{\left(f_1 - f2\widehat{p}_f\left(1 - \widehat{p}_f\right)\right)^2}{f2\widehat{p}_f\left(1 - \widehat{p}_f\right)} + \frac{\left(f_0 - f\widehat{p}_f^2\right)^2}{f\widehat{p}_f^2},$$ (8)

where under the null of HWE, $T_{\text{HWE,female}}$ is asymptotically $\chi_1^2$ distributed.

We first note that earlier work [62], has shown that the Pearson's chi-square-based HWD test can be reformulated as a test based on the HWD estimate, and more recent work [66] has shown that the same HWD test can be derived from a robust allele-based reverse regression. That is, $T_{\text{HWE,female}}$ in Eq (7) can be rewritten equivalently as

$$T_{\text{HWE, female}} = \frac{\widehat{\delta}_f^2}{\frac{1}{f}\widehat{p}_f^2\left(1 - \widehat{p}_f\right)^2}.$$ (9)

We also note that, although the expressions for $\widehat{p}_f^2$ and $\widehat{\delta}_f^2$ in Eq (8) are the same as those in Eqs (3) and (4), here $\widehat{p}_f^2$ and $\widehat{\delta}_f^2$ are calculated separately for each of the five super-populations, as HWD testing in the whole sample using the naïve population-pooled HWD estimate is not valid.

For the four SNPs in the PAR1 and PAR2 regions, HWD estimation and testing were also performed in males, as well as jointly with females using sex-pooled estimates. Briefly, we use $\widehat{\delta}_m$ in Eq (6) to estimate HWD and use

$$T_{\text{HWE, male}} = \frac{\widehat{\delta}_m^2}{\frac{1}{m}\widehat{p}_m^2(1 - \widehat{p}_m)^2},$$ (10)

to test for HWD in males, separately for each of the five super-populations. For the sex-combined analysis,

$$T_{\text{HWE, sex-combined}} = \frac{\widehat{\delta}^2}{\frac{1}{n}\widehat{p}^2(1 - \widehat{p})^2},$$ (11)

where

$$\widehat{p} = \frac{2n_2 + n_1}{2n}, \quad \widehat{\delta} = \widehat{p}(AA) - \widehat{p}^2 = \frac{n_2}{n} - \left(\frac{2n_2 + n_1}{2n}\right)^2, \tag{12}$$

using the notations in S4B Table for a bi-allelic SNP in the PAR1 and PAR2 regions.

To make the HWD test robust to low genotype counts we additionally performed a HWD exact test [25–27] for SNPs in each of the five super-populations, which accommodates both females and males for SNPs from the PAR regions.

## Analyses of autosomal 1, 7 and 22 phase 3 data for benchmarking

To compare results of variants with significant sex difference in MAF between the X chromosome and autosomes, chromosomes 1, 7, and 22 were first selected to represent the longest, similar size to the X chromosome, and one of the shortest autosomes. Biallelic and common (sex- and population-pooled MAF≥5%) SNPs were then selected for sdMAF analysis, using $T_A$, the sdMAF test statistic shown in Eq (5). When HWE evaluation was warranted, $T_{\text{HWE,female}}$, $T_{\text{HWE,male}}$ and $T_{\text{HWE,sex-combined}}$, shown respectively in Eqs (8), (9) and (10), were applied to the phase 3 data of the 1000 Genomes Project.

## Sliding window approach

To better define regions with sdMAF we performed a simple sliding window analysis using a window size of 50 consecutive SNPs, shifted by 25 SNPs each time, and calculated the average of the -log10 p value.

## The X chromosome high-coverage sequence data of the 1000 Genomes Project

To validate the results from the phase 3 data (GRCh37) of the 1000 Genomes Project, we repeated the sdMAF analyses using the recently released high coverage (GRCh38) whole genome sequence data [67].

Specifically, 50, 10, 20, and 50 SNPs, respectively from NPR, PAR1, PAR2, and PAR3, with the smallest sdMAF p-values in the phase 3 data were first selected. Among these SNPs, 4, 10, 10, and 9 SNPs, respectively from NPR, PAR1, PAR2 and PAR3, were successfully lifted-over, bi-allelic in the high coverage data, had no missingness in both sets of data. For each of the 33 SNPs, the direction and magnitude of sdMAF were examined, separately for the phase 3 and high coverage data. Genotype agreements between the two sets of data within an individual, separately by sex, were also generated.

Finally, without the liftover constraint between the two phases of the 1000 Genomes Project, we performed an X-chromosome wide sdMAF analysis for the high coverage data using the same sdMAF methods described earlier for the phase 3 data. Supplementary codes for all the analytical steps are also provided (Web Resource).

## gnomAD v.3.1.2

To examine sdMAF in larger samples with high coverage whole genome sequence we used the genotype and allele counts from the genomes chr X sites VCF file from the non-Finnish European population from gnomAD v 3.1.2 [55] which has a maximum of 19,686 females and 14,343 males (Web Resources). To limit analyses to high quality variants we required them to have both MAF≥0.05 and also allele numbers >50,000 (i.e. >93% call rate in NPR). Similar analysis was performed for African/African American population with the same MAF

threshold, where there are a maximum of 11,086 females and 9,658 males, and this was restricted to variants with allele numbers >30,000 (i.e. >94.2% call rate in NPR).

## Web resources

The Genome Reference Consortium: https://www.ncbi.nlm.nih.gov/grc/human

The 1000 Genomes Project: https://www.internationalgenome.org

Phase 3 data of the 1000 Genomes Project: http://ftp.1000genomes.ebi.ac.uk/vol1/ftp/release/20130502/ and the specific vcf file used:

ftp://ftp.1000genomes.ebi.ac.uk/vol1/ftp/release/20130502/ALL.chrX.phase3_shapeit2_mvncall_integrated_v1b.20130502.genotypes.vcf.gz

High coverage phased data of the 1000 Genomes Project: http://ftp.1000genomes.ebi.ac.uk/vol1/ftp/data_collections/1000G_2504_high_coverage/working/20201028_3202_phased/CCDG_14151_B01_GRM_WGS_2020-08-05_chrX.filtered.eagle2-phased.v2.vcf.gz

gnomAD v.3.1.2 allele counts from genome sequence:

gnomAD v.3.1.2: https://gnomad.broadinstitute.org/downloads

chrX sites VCF: 95.6 GiB, MD5: 040080a18046533728fa60800eedcf4b

gnomAD v.2.1 structural variants: (https://gnomad.broadinstitute.org/gene/ENSG00000124343?dataset=gnomad_sv_r2_1)

## Supporting information

**S1 Table. Counts of X chromosomal variants by regions, global MAF, and types from the phase 3 data of the 1000 Genomes Project on GRCh37.** See Fig 1 for the analytical pipeline of the SNP selection.
(TIF)

**S2 Table. Contrasting combined sample (ALL) and super-population-specific analysis from the 1000 Genomes Project phase 3 data on GRCh37.** The numbers of SNPs with genome-wide significant sdMAF in the superpopulation-specific analysis but not in the ALL analysis, stratified by the superpopulations and regions. The nTotal is the non-overlapping total, stratified by the four regions.
(PNG)

**S3 Table. Counts of X chromosomal variants by regions, global MAF, and types from the high coverage data of the 1000 Genomes Project on GRCh38.** See S19 Fig for the analytical pipeline of the SNP selection.
(PNG)

**S4 Table. Notations of genotype counts for a biallelic SNP on the X chromosome in (A) the NPR and PAR3 regions and (B) the PAR1 and PAR2 regions.** * means not applicable.
(PDF)

**S1 Fig. Counts of males and females by population in the 1000 Genomes Project phase 3 data on GRCh37.** The populations are first ordered by the 5 super-populations alphabetically, and then by the total counts within each super-population. A: counts; B: the corresponding proportion of males. The red horizontal line represents 0.5.
(TIFF)

**S2 Fig. QQ plots of the sdMAF p-values of the X chromosome from the 1000 Genomes Project phase 3 data on GRCh37.** Results of bi-allelic SNPs with global MAF ≥5% are shown separately by region, A: NPR; B: PAR1, C: PAR2; D: PAR3. For better visualization p-values < 1e-300 are plotted as 1e-300 (300 on -log10 scale). The red dashed line represents the

line of equality. The corresponding Manhattan plots are in Fig 2 (across the whole X chromosome) and Figs 4, 5 and 6 for PAR1, PAR2 and PAR3, respectively.
(TIFF)

**S3 Fig. Histograms of the sdMAF p-values of the X chromosome from the 1000 Genomes Project phase 3 data on GRCh37.** Results of bi-allelic SNPs with global MAF≥5% are shown separately by region, A: NPR; B: PAR1, C: PAR2; D: PAR3.
(TIF)

**S4 Fig. Manhattan plot for testing for sex difference in MAF across the X chromosome of** *superpopulation AFR* **from the 1000 Genomes Project phase 3 data on GRCh37.** A: sdMAF p-values for bi-allelic SNPs with MAF ≥5% in superpopulation AFR presumed to be of high quality. SNPs in the PAR1, PAR2 and PAR3 regions are plotted in grey, with PAR3 located around 90 Mb. Y-axis is -log10(sdMAF p-values) and p-values >0.1 are plotted as 0.1 (1 on -log10 scale) for better visualization. The dashed red line represents 5e-8 (7.3 on the -log10 scale). B: Female—Male sdMAF for the same SNPs in part A.
(TIFF)

**S5 Fig. Manhattan plot for testing for sex difference in MAF across the X chromosome of** *superpopulation AMR* **from the 1000 Genomes Project phase 3 data on GRCh37.** A: sdMAF p-values for bi-allelic SNPs with MAF ≥5% in superpopulation AMR presumed to be of high quality. SNPs in the PAR1, PAR2 and PAR3 regions are plotted in grey, with PAR3 located around 90 Mb. Y-axis is -log10(sdMAF p-values) and p-values >0.1 are plotted as 0.1 (1 on -log10 scale) for better visualization. The dashed red line represents 5e-8 (7.3 on the -log10 scale). B: Female—Male sdMAF for the same SNPs in part A.
(TIFF)

**S6 Fig. Manhattan plot for testing for sex difference in MAF across the X chromosome of** *superpopulation EAS* **from the 1000 Genomes Project phase 3 data on GRCh37.** A: sdMAF p-values for bi-allelic SNPs with MAF ≥5% in superpopulation EAS presumed to be of high quality. SNPs in the PAR1, PAR2 and PAR3 regions are plotted in grey, with PAR3 located around 90 Mb. Y-axis is -log10(sdMAF p-values) and p-values >0.1 are plotted as 0.1 (1 on -log10 scale) for better visualization. The dashed red line represents 5e-8 (7.3 on the -log10 scale). B: Female—Male sdMAF for the same SNPs in part A.
(TIFF)

**S7 Fig. Manhattan plot for testing for sex difference in MAF across the X chromosome of** *superpopulation EUR* **from the 1000 Genomes Project phase 3 data on GRCh37.** A: sdMAF p-values for bi-allelic SNPs with MAF ≥5% in superpopulation EUR presumed to be of high quality. SNPs in the PAR1, PAR2 and PAR3 regions are plotted in grey, with PAR3 located around 90 Mb. Y-axis is -log10(sdMAF p-values) and p-values >0.1 are plotted as 0.1 (1 on -log10 scale) for better visualization. The dashed red line represents 5e-8 (7.3 on the -log10 scale). B: Female—Male sdMAF for the same SNPs in part A.
(TIFF)

**S8 Fig. Manhattan plot for testing for sex difference in MAF across the X chromosome of** *superpopulation SAS* **from the 1000 Genomes Project phase 3 data on GRCh37.** A: sdMAF p-values for bi-allelic SNPs with MAF ≥5% in superpopulation SAS presumed to be of high quality. SNPs in the PAR1, PAR2 and PAR3 regions are plotted in grey, with PAR3 located around 90 Mb. Y-axis is -log10(sdMAF p-values) and p-values >0.1 are plotted as 0.1 (1 on -log10 scale) for better visualization. The dashed red line represents 5e-8 (7.3 on

the -log10 scale). B: Female—Male sdMAF for the same SNPs in part A.
(TIFF)

**S9 Fig. Matrix of pairwise scatter plots for comparing sdMAF p-values between combined sample (ALL) and each of the five superpopulations from the 1000 Genomes Project phase 3 data on GRCh37.** The bi-alleic SNPs shown are the ones with global MAF $\geq$5% in the combined ALL sample, polymorphic in each of the five superpopulations, and genome-wide significant in at least one of the six sdMAF analyses (i.e. in ALL or any of the five superpopulations). Both X-axis and Y-axis are -log10(sdMAF p-values) and log-scaled for better visualization. The dashed line is the main diagonal line. Each dot is colored based on the position of the corresponding SNP, with SNPs from PAR1, PAR2, PAR3, and NPR regions colored blue, yellow, red, and black, respectively.
(TIFF)

**S10 Fig. Matrix of pairwise scatter plots for comparing sdMAF between combined sample (ALL) and each of the five superpopulations from the 1000 Genomes Project phase 3 data on GRCh37.** The bi-alleic SNPs shown are the ones with global MAF $\geq$5% in the combined ALL sample, polymorphic in each of the five superpopulations, and genome-wide significant in at least one of the six sdMAF analyses (i.e. in ALL or any of the five superpopulations). Both X-axis and Y-axis are sdMAF. The dashed line represents locations where X and Y have the same sdMAF. Two solid grey lines represent locations where sdMAF from either group are zeros. Each dot in the scatter plots is colored based on the position of the corresponding SNP, with SNPs from PAR1, PAR2, PAR3 and NPR regions colored blue, yellow, red and black respectively.
(TIFF)

**S11 Fig. Miami plot for sdMAF p-values obtained from mega-analysis (top) and meta-analysis (bottom) across the X chromosome from the 1000 Genomes Project phase 3 data on GRCh37.** The bi-alleic SNPs shown are the ones with global MAF $\geq$5% in the combined sample, and polymorphic in each of the five superpopulations (for meta-analysis to have consistent sample sizes across SNPs). SNPs in the PAR1, PAR2 and PAR3 regions are plotted in grey, with PAR3 located around 90 Mb. Y-axis is -log10(sdMAF p-values) and p-values >0.1 are plotted as 0.1 (1 on -log10 scale) for better visualization. The dashed red line represents 5e-8 (7.3 on the -log10 scale). For meta-analysis, a SNP's allele can be sex-specific in a superpopulation. For example, A1 is present only in females, and A2 is present only in males (MAF could be still greater than 5%). In that case, the sdMAF test statistic would be infinity due to variance being zero in the superpopulation sdMAF analysis, which leads to meta-analysis p-value being zero. In those cases, the p-values are plotted as 47,982.36 on the–log10 scale, the smallest p-value observed in the mega-analysis.
(TIFF)

**S12 Fig. Scatter plot contrasting sdMAF p-values obtained from meta-analysis (Y-axis) and mega-analysis (X-axis) across the X chromosome from the 1000 Genomes Project phase 3 data on GRCh37.** The bi-alleic SNPs shown are the ones with global MAF $\geq$5% in the combined ALL sample, polymorphic in each of the five superpopulations (for meta-analysis to have consistent sample sizes across SNPs), and genome-wide significant in either meta-analysis or mega-analysis, or both. Both X-axis and Y-axis are -log10(sdMAF p-values) and log-scaled for better visualization. The dashed line is the main diagonal line. Each dot in the scatter plots is colored based on the position of the corresponding SNP, with SNPs from PAR1, PAR2, PAR3, and NPR regions colored blue, yellow, red, and black, respectively. For meta-analysis, a SNP's allele can be sex-specific in a superpopulation. For example, A1 is present

only in females, and A2 is present only in males (MAF could be still greater than 5%). In that case, the sdMAF test statistic would be infinity due to variance being zero in the superpopulation sdMAF analysis, which leads to meta-analysis p-value being zero. In those cases, the p-values are plotted as 47,982.36 on the–log10 scale, the smallest p-value observed in the mega-analysis.
(TIFF)

**S13 Fig. Bland-Altman plots for X chromosomal SNPs** *with different minor alleles between females and males* **from the 1000 Genomes Project phase 3 data on GRCh37.** Regions are plotted separately A: NPR; B: PAR1, C: PAR2; D: PAR3. For each of the four regions, the histogram at the top of the Bland-Altman plot shows the distribution of the sex-combined MAF for bi-allelic SNPs with global MAF ≥5% presumed to be of high quality. The histogram to the right of the plot shows the distribution of the Female—Male sdMAF. The red dotted lines are the theoretical bounds; see S3 Note for deviations.
(TIFF)

**S14 Fig. Manhattan plot of sdMAF** *sliding window* **p-values across the X chromosome from the 1000 Genomes Project phase 3 data on GRCh37.** Each window contains 50 adjacent bi-allelic SNPs (with global MAF ≥5%) and are moved by 25 SNPs each time. A sliding window sdMAF p-value (on the–log10 scale) is the average of–log10 p-value of the 50 SNPs in the window. The position of each window is represented by the position of the leftmost SNP. SNPs in the PAR1, PAR2 and PAR3 regions are plotted in grey, with PAR3 located around 90 Mb. Y-axis is -log10(sdMAF p-values) and p-values >0.1 are plotted as 0.1 (1 on -log10 scale) for better visualization. The dashed red line represents 5e-8 (7.3 on the -log10 scale).
(TIFF)

**S15 Fig. Manhattan plot for testing for sdMAF across chromosome 1 from the 1000 Genomes Project phase 3 data on GRCh37.** A: sdMAF p-values for bi-allelic SNPs with global MAF ≥5% presumed to be of high quality. Y-axis is −log10(sdMAF p-values) and p-values >0.1 are plotted as 0.1 (1 on −log10 scale) for better visualization. The dashed red line represents 5e-8 (7.3 on the −log10 scale). B: Female—Male sdMAF for the same SNPs in part A.
(TIFF)

**S16 Fig. Manhattan plot for testing for sdMAF across chromosome 7 from the 1000 Genomes Project phase 3 data on GRCh37.** A: sdMAF p-values for bi-allelic SNPs with global MAF ≥5% presumed to be of high quality. Y-axis is −log10(sdMAF p-values) and p-values >0.1 are plotted as 0.1 (1 on −log10 scale) for better visualization. The dashed red line represents 5e-8 (7.3 on the −log10 scale). B: Female—Male sdMAF for the same SNPs in part A.
(TIFF)

**S17 Fig. Manhattan plot for testing for sdMAF across chromosome 22 from the 1000 Genomes Project phase 3 data on GRCh37.** A: sdMAF p-values for bi-allelic SNPs with global MAF ≥5% presumed to be of high quality. Y-axis is −log10(sdMAF p-values) and p-values >0.1 are plotted as 0.1 (1 on −log10 scale) for better visualization. The dashed red line represents 5e-8 (7.3 on the −log10 scale). B: Female—Male sdMAF for the same SNPs in part A.
(TIFF)

**S18 Fig. Histograms and QQ plots of the sdMAF p values for chromosomes 1, 7 and 22 from the 1000 Genomes Project phase 3 data on GRCh37.** Results of bi-allelic SNPs with global MAF ≥5% are shown separately by chromosome: A; chromosome 1; B: chromosome 7; C: chromosome 22. Unlike the X chromosome results in S2 Fig, there was no truncation of sdMAF p-values at 1e-300 as the smallest sdMAF p-value is around 1e-25 for any of these

three autosomes. The red dashed line represents the line of equality.
(TIFF)

**S19 Fig. Pipeline for selection of X chromosomal biallelic SNPs with global MAF ≥5%, presumed to be of high quality, from the 1000 Genomes Project high coverage sequence data on GRCh38.** Variants were placed into the NPR, PAR1, PAR2, and PAR3 regions based on positions available from The Genome Reference Consortium and (19). For detailed counts of variant types and global MAF by regions, see S3 Table.
(PDF)

**S20 Fig. Manhattan plot for testing for sex difference in MAF across the X chromosome from the 1000 Genomes Project high coverage sequence data on GRCh38.** A: sdMAF p-values for bi-allelic SNPs with global MAF ≥5% presumed to be of high quality. SNPs in the PAR1 and PAR3 regions are plotted in grey, with PAR3 located around 90 Mb. Y-axis is −log10(sdMAF p-values) and p-values >0.1 are plotted as 0.1 (1 on −log10 scale) for better visualization. The dashed red line represents 5e-8 (7.3 on the −log10 scale). B: Female—Male sdMAF for the same SNPs in part A. For Zoomed-in plots for the PAR1, PAR2 and PAR3 regions see S21, S22 and S23 Figs, respectively.
(TIFF)

**S21 Fig. Zoomed-in plot for testing for sex difference in MAF across PAR1 of the X chromosome from the 1000 Genomes Project high coverage sequence data on GRCh38.** A: sdMAF p-values for bi-allelic SNPs with global MAF ≥5% presumed to be of high quality. Y-axis is −log10(sdMAF p-values) and p-values >0.1 are plotted as 0.1 (1 on −log10 scale) for better visualization. The dashed red line represents 5e-8 (7.3 on the −log10 scale). B: Female—Male sdMAF for the same SNPs in part A.
(TIFF)

**S22 Fig. Zoomed-in plot for testing for sex difference in MAF across PAR2 of the X chromosome from the 1000 Genomes Project high coverage sequence data on GRCh38.** A: sdMAF p-values for bi-allelic SNPs with global MAF ≥5% presumed to be of high quality. Y-axis is −log10(sdMAF p-values) and p-values >0.1 are plotted as 0.1 (1 on −log10 scale) for better visualization. The dashed red line represents 5e-8 (7.3 on the −log10 scale). B: Female—Male sdMAF for the same SNPs in part A.
(TIFF)

**S23 Fig. Zoomed-in plot for testing for sex difference in MAF across PAR3 of the X chromosome from the 1000 Genomes Project high coverage sequence data on GRCh38.** A: sdMAF p-values for bi-allelic SNPs with global MAF ≥5% presumed to be of high quality. Y-axis is −log10(sdMAF p-values) and p-values >0.1 are plotted as 0.1 (1 on −log10 scale) for better visualization. The dashed red line represents 5e-8 (7.3 on the −log10 scale). B: Female—Male sdMAF for the same SNPs in part A.
(TIFF)

**S24 Fig. QQ plots of the sdMAF p-values of the X chromosome from the 1000 Genomes Project high coverage sequence data on GRCh38.** Results of bi-allelic SNPs with global MAF ≥5% are shown separately by region, A: NPR; B: PAR1, C: PAR2; D: PAR3. For better visualization p-values < 1e-300 are plotted as 1e-300 (300 on -log10 scale). The red dashed line represents the line of equality. The corresponding Manhattan plots are in S20 Fig (across the whole X chromosome) and S21, S22 and S23 Figs for PAR1, PAR2 and PAR3, respectively.
(TIFF)

**S25 Fig. Histograms of the sdMAF p-values of the X chromosome from the 1000 Genomes Project high coverage sequence data on GRCh38.** Results of bi-allelic SNPs with global MAF ≥5% are shown separately by region, A: NPR; B: PAR1, C: PAR2; D: PAR3. The corresponding Manhattan plots are in S20 Fig (across the whole X chromosome) and S21, S22 and S23 Figs for PAR1, PAR2 and PAR3, respectively.
(TIFF)

**S26 Fig. Pipeline for selection of X chromosome biallelic SNPs from the Non-Finnish European population from the gnomAD v3.1.2 data on GRCh38.** Variants were placed into the NPR, PAR1, PAR2, and PAR3 regions based on positions available from The Genome Reference Consortium and (19).
(PDF)

**S27 Fig. Pipeline for selection of X chromosome biallelic SNPs from the African/African American population from the gnomAD v3.1.2 data on GRCh38.** Variants were placed into the NPR, PAR1, PAR2, and PAR3 regions based on positions available from The Genome Reference Consortium and (19).
(PDF)

**S28 Fig. Manhattan plot for testing for sex difference in MAF, for SNPs with 0.1%< MAF<5%, across the X chromosome of the Non-Finnish European population from the gnomAD v3.1.2 data on GRCh38.** A: sdMAF p-values for bi-allelic SNPs with 0.1%< MAF<5%. SNPs in the PAR1, PAR2 and PAR3 regions are plotted in grey, with PAR3 located around 90 Mb. Y-axis is -log10(sdMAF p-values) and p-values >0.1 are plotted as 0.1 (1 on -log10 scale) for better visualization. The dashed red line represents 5e-8 (7.3 on the -log10 scale). B: Female—Male sdMAF for the same SNPs in part A.
(TIFF)

**S1 Note. Bias in (naïve) population-pooled sample estimate of Hardy-Weinberg disequilibrium (HWD) across multiple populations.**
(PDF)

**S2 Note. Comparison of genotypes between phase 3 (GRCh37) and high coverage sequence data (GRCh38) from the 1000 Genomes Project for 23 SNPs selected from the X chromosome.** In total, 50, 10, 20, and 50 SNPs, respectively from NPR, PAR1, PAR2, and PAR3, with the smallest sdMAF p-values in the phase 3 data were first selected. Among these SNPs, 4, 10, 10, and 9 SNPs, respectively from NPR, PAR1, PAR2, and PAR3, were also bi-allelic in the high coverage data and had no missingness in both sets of data. Each page represents the results for one SNP, and SNPs are ordered by the GRCh37 positions. Within each page, the position of the SNP in phase 3 (build GRCh37) and high coverage (GRCh38) are first provided. Next is the female—male sdMAF difference and the sdMAF p-value. The REF and ALT alleles are also provided for each build. Finally, the counts of the agreement of the genotype calls between the phase 3 and the high coverage data are provided, separately by sex.
(PDF)

**S3 Note. Derivations of the theoretical bounds shown in the Bland-Altman plot for X chromosomal SNPs with different minor alleles between females and males.**
(PDF)

**S1 Data. The 1000 Genomes Project phase 3 data and in-depth analysis results for the eight SNPs selected from the X chromosome, and six SNPs selected from the autosomes.** For each of the four regions (NPR, PAR1, PAR2, and PAR3) of the X chromosome and for each

of the three autosomes analyzed (chromosomes 1, 7 and 22), two SNPs with the smallest sdMAF p-values in the 1000 Genomes Project population-combined sample were selected. SNPs are ordered based on GRCh37 position. For each SNP, there are 18 rows, showing the genotype counts and other information for female (F), male (M) and sex-combined (Both) sub-samples for the sex- and population-combined sample (ALL), and the EAS, EUR, AFR, AMR, and SAS super-populations. Thus, there are 18*6 = 108 rows for the six autosomal SNPs, followed by 18*8 = 144 rows for the eight X chromosomal SNPs. Column A: row index 1–252. Column B = SNP: rs name or. if not available from GRCh37. Column C = CHR: 1, 7, 22, or X. Column D = POS: GRCh37 base pair position. Column E = REGION: NA for an autosomal SNP, and NPR, PAR1, PAR2, or PAR3; SNPs are ordered based on GRCh37 position. Column F = A1: the A1 allele defined by PLINK; may not be the minor allele in the sex- and population-pooled ALL sample. Column G = A2: the A2 allele. Column H = Superpopulation: ALL (the sex- and population-combined sample) and the EAS, EUR, AFR, AMR, and SAS super-populations. Column I = Sex: Both (sex-combined), F (female) or M (male). Column J = A1A1 count: genotype count of homozygous A1A1; 0 means zero counts; NA means not applicable for certain cells. For a NPR or PAR3 SNP, the M counts of A1A2 are NA as a result of no heterozygous males, and the sex-combined Both counts of A1A1 and A2A2 are also NA due to the X-inactivation uncertainty. Column K = A1A2 count: genotype count of heterozygous A1A2; Column L = A2A2 count: genotype count of homozygous A2A2. Column M = A1 count: allele count of allele A1; for a SNP in NPR and PAR3 each male only contributes a single allele count. Column N = A2 count: allele count of allele A2; for a SNP in NPR and PAR3 each male only contributes a single allele count. Column O = AF of A1: allele frequency of A1. Column P = AF of A2: allele frequency of A2. Column Q = MA: the minor allele defined based on the sex- and population-pooled ALL sample; NA in other cells. Column R = HWD.delta: the estimate of delta, the measure of HWD which is freq(A1A1)—freq(A1)* freq(A1); NA for population-pooled ALL sample or the male sample when analyzing a SNP in NPR and PAR3. Column S = HWE.p: p-values of HWE testing; NA if HWD.delta is NA or the SNP is monomorphic in that sample. Column T = sdMAF: female—male sex difference in MAF, where the minor allele is defined based on the sex- and population-pooled ALL sample. Column U = sdMAF.p: p-values of sdMAF testing. Column V = HWexact.p: exact p value of HWE testing
(XLS)

**S2 Data. The 1000 Genomes Project phase 3 data and in-depth analysis results for all 2,039 X chromosomal SNPs with genome-wide significant sdMAF.** See legend to S1 Data for other details.
(XLSX)

**S3 Data. Homologous genes in the non-PAR regions of both X and Y, using biomart from ensembl build 38.**
(XLSX)

**S4 Data. The 1000 Genomes Project phase 3 data results for the 52 X chromosomal SNPs with significant sdMAF p-values that are located in X/Y homolog genes.** SNPs are ordered based on GRCh37 position. The template of this table is identical to that of S1 Data
(XLSX)

**S5 Data. The 1000 Genomes Project phase 3 data results for 8 X chromosomal SNPs with the most significant sdMAF p-values in ALL analysis (2 from each region): results separately by 26 populations.** SNPs are ordered based on GRCh37 position. The template of this table is identical to that of S1 Data. These data are plotted in Fig 3.
(CSV)

## Author Contributions

**Conceptualization:** Lei Sun, Andrew D. Paterson.

**Formal analysis:** Zhong Wang.

**Funding acquisition:** Lei Sun, Andrew D. Paterson.

**Investigation:** Zhong Wang, Andrew D. Paterson.

**Methodology:** Zhong Wang, Lei Sun, Andrew D. Paterson.

**Supervision:** Lei Sun, Andrew D. Paterson.

**Visualization:** Zhong Wang.

**Writing – original draft:** Lei Sun, Andrew D. Paterson.

**Writing – review & editing:** Zhong Wang.

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
