## [Decision Letter · Decision Letter 0]

12 Jan 2022

Dear Dr Paterson,

Thank you very much for submitting your Research Article entitled 'Major sex differences in allele frequencies for X chromosome variants in the 1000 Genomes Project data' to PLOS Genetics.

The manuscript was fully evaluated at the editorial level and by independent peer reviewers. The reviewers appreciated the attention to an important problem, but raised some substantial concerns about the current manuscript. Based on the reviews, we will not be able to accept this version of the manuscript, but we would be willing to review a much-revised version. We cannot, of course, promise publication at that time.

Should you decide to revise the manuscript for further consideration here, your revisions should address the specific points made by each reviewer. Importantly, the issue of population structure should be addressed carefully as pointed out by Reviewer 3. We will also require a detailed list of your responses to the review comments and a description of the changes you have made in the manuscript. 

If you decide to revise the manuscript for further consideration at PLOS Genetics, please aim to resubmit within the next 60 days, unless it will take extra time to address the concerns of the reviewers, in which case we would appreciate an expected resubmission date by email to plosgenetics@plos.org.

[LINK]

We are sorry that we cannot be more positive about your manuscript at this stage. Please do not hesitate to contact us if you have any concerns or questions.

Yours sincerely,

Heather J Cordell

Associate Editor

PLOS Genetics

Hua Tang

Section Editor: Natural Variation

PLOS Genetics

Reviewer's Responses to Questions

**Comments to the Authors:**

Reviewer #1: The authors report X-chromosome allele frequency differences in 1000 Genomes data. The data are of interest, but the statistical analyses do not take into account the methodology of Graffelman (Heredity 116:558-568, 2016; Genetic Epidemiology 42:34-48, 2018).

Reviewer #2: General comments:

In the manuscript ‘Major sex differences in allele frequencies for X chromosome variants in the 1000

Genomes Project data’, the authors examine sex differences in minor allele frequency on the X chromosome using 1000 genomes project data, focusing on the different X chromosome regions, comparisons between low and high coverage sequencing, and comparisons to the autosomes. They observe sex differences in allele frequency in PAR1 and PAR2, which may point to underlying biology and differences likely due to linkage, whereas differences in the NPR and PAR3 are likely due to genotyping errors in lower coverage data. The paper is extremely well-written and clear, and represents an important contribution, considering that the X chromosome is often excluded from analyses because of a lack of understanding of the complex issues and appropriate resources and analysis tools. These results have implications for how X chromosome data should be analyzed and interpreted, and notably also points to interesting biology that may underly the observed differences in variant frequencies between males and females. Specific observations and clarifications are outlined below.

Specific comments:

• The investigation breaks out SNPs in region, between non-PAR and PAR. But what about SNPs in the non-PAR region that reside within genes that have homologous copies on the Y? What do the allele frequencies look like for these SNPs on both the X and the Y in males? Similarly, what do the allele frequencies look like on the Y more generally? Perhaps this would also serve as an appropriate comparison, especially for the PAR region.

• How big are the observed sex differences in MAF? The focus is mostly on p-values, but what are the general distributions of the effect size differences? These are shown in figures, but only described for highly significant variants.

• The in-depth analysis of 8 SNPs shows some interesting patterns. How wide-spread are these across the significant variants? For example, minor allele flips between males and females, and the fixed homozygous genotype in females but fixed heterozygous in males (or mix of homozygous and heterozygous in the less extreme case). This suggest that one allele is fixed on the X and the other allele is fixed on the Y. How is it possible in the PAR with recombination between X and Y? Because it is an A/T, is this actually due to an error/strand flip issue? Or is the recombination rate 0 in this region?

• Why do we see higher rates of allele frequency differences in PAR2 compared to other regions, particularly PAR1? Is this in line with the higher recombination rate (overall and sex-specific) observed in PAR1 in Monteiro et al?

• Why weren’t all SNPs in phase 3 examined in high-coverage data? Does the large lift-over failure rate point to systematic errors in the successful SNPs, or only the unsuccessful? Was the lift-over failure rate the same across the four regions?

• Did you consider a sliding window approach to compute allele frequencies and allele frequency differences? This may allow for a better regional description of patterns.

• Are similar patterns seen in other datasets, e.g. HRC and TopMed? And are sex differences in allele frequency seen with rare variants?

• It would be helpful to emphasize more the implications of these findings for analysis methods, especially for the PAR genes. For example, differences in allele frequency will lead to variance differences. What are the solutions and recommendations, and do they differ between PAR and non-PAR? Are sex-specific results necessary? Findings also seem to suggest that sex-specific QC is critical for the X (MAF, genotyping error, and HWE). Due to sex-specific differences in HWE, should this be part of QC?

• A paragraph on next steps, limitations, and/or conclusion

Reviewer #3: Review of “Major sex differences in allele frequencies for X chromosome variants in the 1000 Genomes Project data” by Wang et al.

This article reports the interesting finding that a set of SNPs on the X chromosome, in particular those close to the boundaries of the PAR regions, shows significant differences in allele frequency between males and females. The article has extensive supplementary graphics, reflecting results of considerable amount of data analysis performed with the samples from the 1000G project. I have some major and minor concerns detailed below.

Major concerns:

In the introduction the author sketch that the X chromosome has been largely ignored in association studies, and that methods to deal with X chromosomal data have only recently been developed. This is certainly true, but not only so for association studies. In fact, even methods for testing for Hardy-Weinberg equilibrium at X chromosomal variants are very new (Graffelman & Weir, 2016). To my eyes the Introduction section of the paper is somewhat short, and a bit more background on the X chromosome (Wise et al., 2013) and the PAR regions (Graves et al., 1998) could strengthen the article.

The initial analysis on detecting sdMAF SNPs is based on the full 1000G data. This comprises individuals from 26 human populations and five continental regions worldwide. It is well known that allele frequencies of SNPs tend to vary over human populations, and that ignoring such population substructure can lead to spurious statistical associations in many types of analysis: Hardy-Weinberg equilibrium, linkage disequilibrium (LD) and association testing among others. In the Online methods section, the authors state in this regard about their sdMAF SNP test procedure: “This more powerful approach (i.e. by joining the populations) can detect sex difference in MAF if it is present in any of the sub-samples”. This is not quite true. If two populations both have a sex difference in MAF, but of opposite sign (e.g. population 1 having A as the minor allele, and population 2 having B as the minor allele) then joining those two populations averages out the difference in MAF between the sexes. This shows that joining samples from different populations is not always beneficial, but can lead to a loss of power to detect sdMAF SNPs. In order to avoid the effects of population substructure, one could just try to detect sdMAF SNPs WITHIN each population. That would probably give a fairer idea of how many of such sdMAF SNPs are around, and it could detect additional sdMAF SNPs that have may have gone unnoticed in the current analysis.

The authors use a fixed genome-wide significance level of 5E-8 throughout, for all chromosomes studied, but the question arises if this is adequate. They find 6, 1 and 0 sdMAF SNPs on the autosomes 1, 7 and 22 respectively, but such decay is to be expected, simply because the larger chromosomes have more SNPs, and thus also more significant ones. Whether there is truly a different rate of significant variants across the chromosomes is probably better judged by the comparing the number of significant results after a Bonferroni correction for each chromosome, since this will take the number of SNPs on each chromosome into account.

The authors conduct tests for HWE of X chromosomal variants. They use the standard autosomal asymptotic chi-square test for the NPR and PAR3 region, using the females only. This is not to be recommend for various reasons. First of all, the HWE exact test is the state-of-the-art statistical methodology for testing HWE, for having in general, a better type 1 error rate and power (Wigginton et al., 2005; Graffelman & Moreno, 2013). Moreover, by running the test on females only, the male X chromosomal alleles are ignored, leading to loss of power, and, importantly, if male and female allele frequencies differ HWE will not hold due to the fact that convergence to equilibrium can take several generations for the X chromosome (Crow & Kimura, 1970), and this can go undetected by testing females only. The best way to test the NPR (and presumably, the PAR3 region, which is not really established as pseudo-autosomal) is to use the X chromosomal exact test for HWE, specially developed for this purpose by Graffelman & Weir (2016). For testing variants in PAR1, the authors report p-values can be “NA” for monomorphic variants. This again refers to asymptotic testing with the Chi-square test. The exact test for HWE can deal with monomorphic variants, and yields a p-value of 1 for monomorphic variants, which looks more neat. For PAR1 and PAR2, the natural thing is to use the standard autosomal exact test for HWE for all observations (males and females), and given the focus of the paper on sdMAF SNPs, one might consider to do the exact test also for males and females separately. However, it has shown that tests for equal allele frequencies in the sexes and for HWE are intricately linked in their assumptions (Graffelman & Weir, 2018), and therefore, for PAR1 and PAR2 the joint exact test for equality of allele frequencies and Hardy-Weinberg proportions is indicated. The forgoing comments show that the “new X chromosome-aware HWE methods that consider sdMAFs” the authors call for do in fact already exist but they were apparently not noticed by the authors.

Minor issues:

“X chromosome variants” in the title probably better worded as “X chromosomal variants”.

I suggest to replace 245825 SNPs by 245825 X-chromosomal SNPs to stress the number is specific for the X chromosome.

References:

Crow, J.F. and Kimura, M. (1970) An introduction to population genetics theory. Harper & Row, publishers, New York.

Graffelman, J. and Weir, B. S. (2016) Testing for Hardy-Weinberg equilibrium at bi-allelic genetic markers on the X chromosome. Heredity 116 6:558—568. Doi: 10.1038/hdy.2016.20

Graffelman, J. and Weir, B.S. (2018) On the testing of Hardy-Weinberg proportions and equality of allele frequencies in males and females at bi-allelic genetic markers. Genetic Epidemiology 42(1):34—48. Doi: 10.1002/gepi.22079

Graffelman, J. and Moreno, V. (2013) The mid p-value in exact tests for Hardy-Weinberg equilibrium. Statistical Applications in Genetics and Molecular Biology. 12(4), pp. 433--448. doi: 10.1515/sagmb-2012-0039

Graves JAM, Wakefield MJ, Toder R (1998). The origin and evolution of the pseudoautosomal regions of human sex chromosomes. Human Molecular Genetics 7: 1991–1996. Doi: 10.1093/hmg/7.13.1991

Wigginton, J. E. and Cutler, D. J. and Abecasis, G. R. (2005) A note on exact tests of Hardy-Weinberg equilibrium American Journal of Human Genetics 76: 887-893. Doi: 10.1086/429864

Wise AL, Gyi L, Manolio TA. (2013) eXclusion: toward integrating the X chromosome in genome-wide association analyses. American Journal of Human Genetics. 92(5):643-647. Doi: 10.1016/j.ajhg.2013.03.017

**Have all data underlying the figures and results presented in the manuscript been provided?**

Reviewer #1: None

Reviewer #2: Yes

Reviewer #3: Yes

PLOS authors have the option to publish the peer review history of their article (what does this mean?). If published, this will include your full peer review and any attached files.

Reviewer #1: No

Reviewer #2: No

Reviewer #3: No

---

## [Decision Letter · Decision Letter 1]

5 Apr 2022

Dear Dr Paterson,

Thank you very much for submitting your Research Article entitled 'Major sex differences in allele frequencies for X chromosomal variants in both the 1000 Genomes Project and gnomAD' to PLOS Genetics.

The manuscript was fully evaluated at the editorial level and by independent peer reviewers. The reviewers appreciated the attention to an important topic but identified some remaining concerns that we ask you address in a revised manuscript

We therefore ask you to modify the manuscript according to the review recommendations. Your revisions should address the specific points made by each reviewer.

[LINK]

Yours sincerely,

Heather J Cordell

Associate Editor

PLOS Genetics

Hua Tang

Section Editor: Human Variation

PLOS Genetics

Reviewer's Responses to Questions

**Comments to the Authors:**

Reviewer #1: The authors have addressed my concerns.

Reviewer #2: My previous concerns have all been addressed. I have no further comments.

Reviewer #3: Re-review of “Major sex differences in allele frequencies for X chromosome variants in the 1000 Genomes Project data” by Wang et al.

This article has improved after the first round of review. However, the authors fail to adequately address several of the issues previously expressed. I have the following remaining major and minor concerns. The comments below refer to the updated manuscript, using line numbers of the new clean version without tracked changes.

Major concerns:

References. The authors claim to “have also added the …. Graves et al., 1998 paper to the references”, however the truth is that they have not cited Graves et al. (1998) in the new version of the manuscript.

L182: with respect to the in-depth analysis of a small set of most significant sdMAF SNPs, the paper may benefit from a table, included in the main manuscript, containing genotype counts and test results for sdMAF and HWE of these SNPs. This avoids the reader has to browse supplementary Excel files in order to see some detailed examples of, after all, the most salient sdMAF SNPs, which are the core topic of this paper.

Population substructure. The way population substructure is currently addressed in the paper leaves still much to desire. In the first place the authors “account for” population substructure by doing separate analyses at the CONTINENTAL level. Allele frequency differences over SNPs within continents are certainly known to exist. The authors never truly go down to testing at the population level. Repeating the analysis for all 26 1000G populations is of course a prodigious amount of work, and not really required, but evidence for the existence of sdMAF SNPs would certainly be more convincing, if all 26 populations are tested for at least some of the most salient sdMAF SNPs that are found by the authors. This will be insightful, because it will also show if sdMAF SNPs can be population specific, or if they are preserved across all populations. Moreover, as pointed out in the first review, testing within each population may detect additional sdMAF SNPs that have may have gone unnoticed in the continental analysis, and this could at least be mentioned.

The limitations of the sdMAF test (such as the mentioned loss of power due to the averaging out of allele frequency differences across populations in the previous review) should be clearly addressed in the Discussion section, in particular because no detailed population-level analysis has been carried out.

Hardy-Weinberg equilibrium testing. On lines 214-219 the authors refer to HWE testing in the PAR region, and their comment that “monomorphic females can have p-value=NA” shows they continue to focus on the chi-square test, whereas exact procedures are known to be more powerful. As already pointed out in my previous review: “The exact test for HWE can deal with monomorphic variants, and yields a p-value of 1 for monomorphic variants, which looks more neat. For PAR1 and PAR2, the natural thing is to use the standard autosomal exact test for HWE”. The authors should clearly mention the aptness of the classical exact test for the PAR regions, and state that monomorphic SNPs will not have p-value NA but p-value 1 in that case.

On lines 415-416 the authors state “Earlier work has examined HWD of X chromosomal variants (58, 59), using a 2 degrees of freedom (df) test that jointly analyses both females and males…”. This sentence is notoriously false. Reference 59, Graffelman et al. (2017), clearly and explicitly state all SNPs (autosomal and X-chromosomal) are tested with EXACT tests, NOT with a 2df chi-square test. The exact tests are to be preferred, because they are the most powerful and state-of-the-art approach to HWE testing. Moreover, Graffelman and Weir (2016) explicitly show the Type 1 error rate of the Exact test for X is better than the 2DF chi-square test for X.

On lines 422-424 the authors state “And for NPR and PAR3 variants, we performed the HWD analysis in females only; the males have two hemizygous genotypes, leaving no df to perform the HWD test after using one df to estimate the MAF in males.” First of all, this is the most common situation on X, as the non-recombining NPR part is the largest, and contains most of the SNPs. Second, what the authors do is to apply the classical one-degree-of-freedom chi-square test for the females only in this case. That is not the best choice of test. There is no need to exclude the male alleles. As explicitly argued by Graffelman & Weir (2016), for NPR (and PAR3) X chromosomal variants, a specific X-chromosomal exact test can be used that uses ALL alleles (males and females), and this test is to be preferred for the reasons given above.

Most notoriously, the authors cite the article of Graffelman & Weir (2016), they analyse X chromosomal NPR/PAR3 data, but the do not even mention that a specific HWE Exact test procedure for NPR and PAR3 variants exists.

Consequently, the paragraph starting at line 415 should be completely revised.

On lines 565-570 the authors again stress the use of asymptotic chi-square-statistic for also doing the testing of HWE in females only in the NPR and PAR3 region. Again, in this case the most powerful test is the X-chromosomal Exact test for HWE; as already literally stated in the previous review “The best way to test the NPR (and presumably, the PAR3 region, which is not really established as pseudo-autosomal) is to use the X chromosomal exact test for HWE, specially developed for this purpose by Graffelman & Weir (2016).”

If the authors desire also to test females only, then they should just do the standard autosomal Exact test with the female genotype counts only. In the literature, this has been shown to be superior to the chi-square test; the exact test better controls the Type I error rate, and has better power (Wigginton, 2005; Graffelman & Moreno, 2013).

L572: the reformulation in Eq. (9) is cited to be “recent work”, but is in fact well known, and much older, and can be found in, among others, Weir (1996).

Minor issues:

L228: “have close” � “has close”

L494: Table S3 seems to refer to table S4.

L512: “consists of” � “consisting of”

L538: omit “with each other”

Equation (9): “r” is not defined, and seems mistaken, and apparently needs to be replaced by “f”.

References:

Graves JAM, Wakefield MJ, Toder R (1998). The origin and evolution of the pseudoautosomal regions of human sex chromosomes. Human Molecular Genetics 7: 1991–1996. Doi: 10.1093/hmg/7.13.1991

Graffelman, J., Jain, D. and Weir, B.S. (2017) A genome-wide study of Hardy-Weinberg equilibrium with next generation sequence data. Human Genetics 136(6) pp. 727-741. doi: 10.1007/s00439-017-1786-7.

Weir, B. S. (1996) Genetic Data Analysis II. Sinauer Associates, Massachusetts.

Wigginton JE, Cutler DJ, Abecasis GR (2005). A note on exact tests of Hardy-Weinberg equilibrium. Am J Hum Genet 76: 887–893.

**Have all data underlying the figures and results presented in the manuscript been provided?**

Reviewer #1: Yes

Reviewer #2: Yes

Reviewer #3: Yes

PLOS authors have the option to publish the peer review history of their article (what does this mean?). If published, this will include your full peer review and any attached files.

Reviewer #1: No

Reviewer #2: No

Reviewer #3: No

---

## [Decision Letter · Decision Letter 2]

3 May 2022

Dear Dr Paterson,

We are pleased to inform you that your manuscript entitled "Major sex differences in allele frequencies for X chromosomal variants in both the 1000 Genomes Project and gnomAD" has been editorially accepted for publication in PLOS Genetics. Congratulations!

Yours sincerely,

Heather J Cordell

Associate Editor

PLOS Genetics

Hua Tang

Section Editor: Human Variation

PLOS Genetics

Comments from the reviewers (if applicable):

Reviewer's Responses to Questions

**Comments to the Authors:**

Reviewer #3: The article has improved after the second round of review. I have no further comments.

**Have all data underlying the figures and results presented in the manuscript been provided?**

Reviewer #3: Yes

PLOS authors have the option to publish the peer review history of their article (what does this mean?). If published, this will include your full peer review and any attached files.

Reviewer #3: No

**Data Deposition**

http://datadryad.org/submit?journalID=pgenetics&manu=PGENETICS-D-21-01453R2

**Press Queries**

---

## [Editor Report · Acceptance letter]

25 May 2022

PGENETICS-D-21-01453R2 

Major sex differences in allele frequencies for X chromosomal variants in both the 1000 Genomes Project and gnomAD 

Dear Dr Paterson, 

We are pleased to inform you that your manuscript entitled "Major sex differences in allele frequencies for X chromosomal variants in both the 1000 Genomes Project and gnomAD" has been formally accepted for publication in PLOS Genetics! Your manuscript is now with our production department and you will be notified of the publication date in due course.

With kind regards,

Livia Horvath

PLOS Genetics

On behalf of:
